# Similarity, Compression and Local Steps: Three Pillars of Efficient Communications for Distributed Variational Inequalities

**Aleksandr Beznosikov**
Innopolis University, Skoltech, MIPT, Yandex

**Martin Takáč**
MBZUAI

**Alexander Gasnikov**
MIPT, Skoltech, IITP RAS

## Abstract

Variational inequalities are a broad and flexible class of problems that includes minimization, saddle point, and fixed point problems as special cases. Therefore, variational inequalities are used in various applications ranging from equilibrium search to adversarial learning. With the increasing size of data and models, today's instances demand parallel and distributed computing for real-world machine learning problems, most of which can be represented as variational inequalities. Meanwhile, most distributed approaches have a significant bottleneck – the cost of communications. The three main techniques to reduce the total number of communication rounds and the cost of one such round are the similarity of local functions, compression of transmitted information, and local updates. In this paper, we combine all these approaches. Such a triple synergy did not exist before for variational inequalities and saddle problems, nor even for minimization problems. The methods presented in this paper have the best theoretical guarantees of communication complexity and are significantly ahead of other methods for distributed variational inequalities. The theoretical results are confirmed by adversarial learning experiments on synthetic and real datasets.

## 1 Introduction

Variational Inequalities (VI) have a rather long history of research. Their first applications were the equilibrium problems in economics and game theory [72, 34, 29]. Variational inequalities were seen as a universal paradigm, involving other attractive problems, including minimization, Saddle Point Problems (SPP), fixed point problems, and others. Whereas research for minimization was actively developed separately, research for saddle point problems was often coupled with science's development around variational inequalities. This trend continues even now. This is partly related to the fact that the methods for minimization problems are not always suitable for SPP and VI. Moreover, from the theoretical point of view, these methods give relatively poor or even zero guarantees of convergence. Therefore a separate branch of algorithms has been developed for VI and SPP. Such methods include, e.g. `Extra Gradient` [50, 71, 42], Tseng's method [84] and many others [14]. Over time, VI found new applications, e.g. in signal processing [77]. In the last decade, they also attracted the interest of the machine learning community. Nowadays, VI and SPP are used in various aspects of machine learning. For example, there are many applications for the already classical learning tasks in supervised learning (with non-separable loss [40]; with non-separable regularizer [6]), unsupervised learning (discriminative clustering [89]; matrix factorization [7]), image denoising [28, 17], robust optimization [10]. It turns out that VI can be also linked with modern areas of learning science, including reinforcement learning [73, 39], adversarial training [62], and GANs [31, 23, 30, 67, 20, 57, 74] to name a few.

Machine learning's rapid development offers more possibilities for consumers, but poses challenges for engineers. To meet modern society's demands, the performance and generalization capacity of the model must be improved. This is currently done by increasing training data and model size [5].

37th Conference on Neural Information Processing Systems (NeurIPS 2023).

Hardware acceleration and most importantly distributed algorithm design are essential for effective handling of these workloads [85]. The distributed nature of the problem occurs not only due to a desire to speed up the learning process via parallelization, but because of the severe external constraints, e.g. as in Federated Learning (FL) [49, 56]. FL is a pioneering area of distributed learning, where users' personal devices are computing machines. Private data transfer from local devices is prohibited due to privacy constraints, making distributed approaches the only practical option. Most distributed algorithms have communications as a bottleneck. Communication wastes time, slowing down training/optimization and preventing effective parallelization. Different approaches can reduce communication rounds and cost per round. The first popular approach is the compression of transmitted information [2]; it helps to increase the speed of the communication round without much loss in the quality of learning. The second approach is use of local steps [64]; by increasing the number of local computations, it is sometimes possible to

Table 1: Summary of the results on the number of transmitted information in different approaches to **communication bottleneck for distributed VI/SPP**.

| Method | Approach | Communication complexity |
|---|---|---|
| Extra Gradient [50, 42] | | $\mathcal{O}\left(\frac{L}{\mu}\log\frac{1}{\varepsilon}\right)$ |
| Local SGDA [25] | local steps | $\mathcal{O}\left(\frac{L^{2/5}n^{2/5}}{\mu^{2/5}\varepsilon^{1/5}}+\frac{L\zeta}{\mu^{3/2}\sqrt{\varepsilon}}\right)$ |
| FedAvg-S [36] | local steps | $\mathcal{O}\left(\frac{L^2}{\mu^2}\log\frac{1}{\varepsilon}+\frac{L\zeta}{\mu^2\sqrt{\varepsilon}}\right)$ |
| SCAFFOLD-S [36] | local steps | $\mathcal{O}\left(\frac{L^2}{\mu^2}\log\frac{1}{\varepsilon}\right)$ |
| SCAFFOLD-Catalyst-S [36] | local steps | $\mathcal{O}\left(\frac{L}{\mu}\log^2\frac{1}{\varepsilon}\right)$ |
| ESTVGM [11] | local steps | $\mathcal{O}\left(\frac{L}{\mu}\log\frac{1}{\varepsilon}+\frac{L\zeta}{\mu^2\sqrt{\varepsilon}}\right)$ |
| SMMDS [16] | similarity local steps | $\mathcal{O}\left(\left[1+\frac{\delta}{\mu}\right]\log\frac{1}{\varepsilon}\right)$ |
| MASHA [15] | compression | $\mathcal{O}\left(\frac{L}{\sqrt{n}\mu}\log\frac{1}{\varepsilon}\right)$ |
| Optimistic MASHA [12] | similarity compression | $\mathcal{O}\left(\left[\frac{L}{n\mu}+\frac{\delta}{\sqrt{n}\mu}\right]\log\frac{1}{\varepsilon}\right)$ |
| Accelerated Extra Gradient [51] | similarity local steps | $\mathcal{O}\left(\left[1+\frac{\delta}{\mu}\right]\log\frac{1}{\varepsilon}\right)$ |
| Algorithm 1 (this paper) | similarity local steps compression | $\mathcal{O}\left(\left[1+\frac{\delta}{\sqrt{n}\mu}\right]\log\frac{1}{\varepsilon}\right)$ |
| Algorithm 2 (this paper) | similarity local steps partial participation | $\mathcal{O}\left(\left[1+\frac{\delta}{\sqrt{n}\mu}\right]\log\frac{1}{\varepsilon}\right)$ |
| Lower bound [16] | | $\Omega\left(\left[1+\frac{\delta}{\mu}\right]\log\frac{1}{\varepsilon}\right)$ [(1)] |
| Lower bound (this paper) | similarity local steps compression | $\Omega\left(\left[1+\frac{\delta}{\sqrt{n}\mu}\right]\log\frac{1}{\varepsilon}\right)$ |
| Lower bound (this paper) | similarity local steps partial participation | $\Omega\left(\left[1+\frac{\delta}{\sqrt{n}\mu}\right]\log\frac{1}{\varepsilon}\right)$ |

[(1)] lower bound is deterministic and does not take into account the possibility of compression and partial participation. *Notation:* $\mu$ = constant of strong monotonicity of the operator $F$ (strong convexity – strong concavity of the corresponding saddle point problem), $L$ = Lipschitz constant of $F$, $\delta$ = similarity parameter (typically, $\delta \ll L$), $\zeta$ = heterogeneity parameter, $n$ = number of devices, $b$ = local data size, $\varepsilon$ = precision of the solution.

significantly reduce the number of communication rounds. Finally, the third basic approach relies on data similarity [79]. Local steps are often employed to help with this approach. Data similarity allows us to assume local loss functions on devices are similar, meaning local computations are close and replaceable. Both approaches and combinations have been extensively researched for minimizations, but a synergy of all three pillars of distributed algorithms has not been presented for either minimization problems or VI. Therefore, the purpose of this paper is to answer the following questions:

*Is it possible to combine the three main techniques for dealing with the comm. bottleneck into one algorithm? Does the new method have better convergence in both theory and practice?*

## 2 Related works

Recent years have seen a surge of research into effective communication techniques for distributed algorithms. Much of this work focuses on minimization problems, but there are also methods for SPP and VI that address the communication bottleneck issue. We summarize these approaches in Table 1.
• **Local methods.** The use of local steps is probably the most popular approach to the communication bottleneck. Methods with local updates are also widely used in general distributed and FL. The idea of distributed minimization using methods similar to Local SGD is not an achievement of recent years and comes back to [64, 65, 91]. The development of theoretical analysis and new modifications for the original methods can be traced in the following works [58, 80, 46, 45, 87, 86, 48]. With the appearance of [69], local methods for distributed minimization problems get a new boost. The first local methods for VI and SPP were simply adaptations of local algorithms for the minimization problem. In particular, the `Local SGD/FedAvg` method was adapted in [25, 36]. [36] also analysed the Scaffold method [45] for SPP. In [11], the `Extra Gradient` method, a key method for SPP and VI, was modified by the use of local steps.
• **Compression.** Methods with compression also have a good background. Here we can highlight

the basic methods that add compression to GD or SGD [2], and modifications of this approach using memory [68]. Also of interest are works investigating biased compression operators [13] and error-compensation techniques [3, 81, 76]. For VI and SPP, methods with different compressions were proposed [15]. These algorithms are based on the variance reduction method. See also [76] and [15] for a more detailed survey of compressed methods for distributed minimization problems.

• **Similarity.** This paper examines the similarity of Hessians in distributed minimization problems, building on established theory and practice (Assumption 4.3). The first method for distributed minimization problems with data similarity was DANE [79]. The lower bounds on communications under similarity assumption was discussed in [4]. The optimal method was obtained in [90] (for quadratic problems) and [51] (for general case). For details and a complete picture of the methods for the distributed minimization problems with similarity, see [51] and [35]. The SPP were also considered under similarity assumption. In [16], both the lower bounds and the optimal method were obtained. In the current work, we "break down" these lower estimates by using compression (see Table 1 with footnote 1). Our current study advances these concepts by integrating compression techniques, significantly outperforming previous estimates (see Table 1, footnote 1). Most related methods (for the distributed problem with similarity) incorporate local steps.

• **Local methods with compression.** The combination of local updates and compression for minimization problems has been presented in [9, 70] (note that the theoretical rates are not optimal and used restrictive assumptions). Using the same local approach as in [69], in papers [63, 21], authors combine it with compression technique and get good convergence rates. For VI and SPP, there are no methods with both local steps and compression.

• **Compression and similarity.** Recently, a new class of compressions called permutation compressors was introduced for the distributed minimization problem under the data similarity assumption [82]. However, due to not using local steps technique, the gain from combining these two approaches was slight. For VI the close results were presented in [12].

## 3 Our contributions

We answer in the affirmative to the questions posed at the end of Section 1. In particular, our contributions are as follows.

• **Three Pillars Algorithm.** We present a new method Three Pillars Algorithm (Algorithm 1) that combines three approaches: compression, similarity, and local steps for the efficient solution of distributed VI and SPP.

• **The best communication complexity.** We analyze the convergence of the new distributed algorithm for smooth strongly monotone VI and strongly convex-strongly concave SPP under similarity assumption. As can be seen in Table 1, our algorithm has better communication complexity than all available competitors. Since $\delta \leq L$, it is easy to see that our algorithms always outperform all methods except Optimistic MASHA [12]. One can note that in the worst case with $\delta \sim L$ we repeat the results of Optimistic MASHA. But if the data are uniformly distributed, then $\delta \sim \frac{L}{\sqrt{b}}$ or even $\frac{L}{b}$, where $b$ is the size of the local sample. Assume that $N = bn$ is the size of the whole data, then our methods outperform Optimistic MASHA when $L > \delta\sqrt{n}$, i.e. when $N > n^2$. Such a ratio is typical because already classic modern datasets contain hundreds of thousands [54, 53] or even millions [24] of samples, and the number of devices rarely exceeds 100.

• **Main calculations on server.** Many distributed approaches, such as one of the most popular from [66], use the power of all computing devices evenly. But in terms of FL, we prefer to have the main computational workload on the server, rather than on the local devices of the users [16, 51]. Our algorithm has this feature.

• **Extension with partial participation.** We present a modified version of our first method – Algorithm 2 (see Appendix A). Instead of compression, one device is selected to send an uncompressed message. This reduces the local complexity on devices compared to Algorithm 1, while keeping the overall communication complexity the same.

• **Lower bounds.** To demonstrate the optimality of our proposed approaches, we establish lower bounds for the communication complexities. These estimates enhance those from [16], which did not consider the potential for compression and partial participation. Our results are shown to be unsurpassable in both scenarios involving compression and partial participation (see Table 1).

• **Extension with stochastic local computations.** Motivated by applications where computation is expensive, we present a modification of Algorithm 1 that emphasizes the use of stochastic operators in local computations, termed Algorithm 3 (see Appendix A).

• **Non-monotone analysis.** Although we focus primarily on strongly monotone VI, many machine learning problems are non-monotone, and hence we extend the convergence analysis to this case. Our

algorithm outperforms competitors in terms of communication complexity for both cases and is the first to gain acceleration in the non-monotone case.

• **Experimental confirmation.** Experimental results on adversarial training of regression models confirm the theoretical conclusion that both of our algorithms outperform their competitors.

## 4   Problem and assumptions

We denote standard inner product of $x, y \in \mathbb{R}^d$ as $\langle x, y \rangle = \sum_{i=1}^d x_i y_i$, where $x_i$ corresponds to the $i$-th component of $x$ in the standard basis in $\mathbb{R}^d$. It induces $\ell_2$-norm in $\mathbb{R}^d$ in the following way $\|x\| = \sqrt{\langle x, x \rangle}$. Operator $\mathbb{E}[\cdot]$ denotes full mathematical expectation and $\mathbb{E}_\xi[\cdot]$ introduces the conditional mathematical expectation on $\xi$.

Formally, the problem of finding a solution for the variation inequality can be stated as follows:
$$\text{Find } z^* \in \mathcal{Z} \text{ such that } \langle F(z^*), z - z^* \rangle \geq 0, \quad \forall z \in \mathcal{Z}, \tag{1}$$
where $F : \mathcal{Z} \to \mathbb{R}^d$ is an operator, and $\mathcal{Z} \subseteq \mathbb{R}^d$ is a convex set. We assume that the training data describing $F$ is distributed across $n$ devices/nodes: $F(z) = \frac{1}{n}\sum_{i=1}^n F_i(z)$, where $F_n : \mathcal{Z} \to \mathbb{R}^d$ for all $i \in [n] := \{1, 2, \ldots, n\}$. To emphasize the extensiveness of the formalism (1), we give some examples of variational inequalities.

• **Minimization.** Consider the minimization problem:
$$\min_{z \in \mathcal{Z}} f(z). \tag{2}$$
Suppose that $F(z) = \nabla f(z)$. Then, if $f$ is convex, it can be proved that $z^* \in \mathcal{Z}$ is a solution for (1) if and only if $z^* \in \mathcal{Z}$ is a solution for (2).

• **Saddle point problem.** Consider the saddle point problem:
$$\min_{x \in \mathcal{X}} \max_{y \in \mathcal{Y}} g(x, y). \tag{3}$$
Suppose that $F(z) = F(x, y) = [\nabla_x g(x, y), -\nabla_y g(x, y)]$ and $\mathcal{Z} = \mathcal{X} \times \mathcal{Y}$ with $\mathcal{X} \subseteq \mathbb{R}^{d_x}$, $\mathcal{Y} \subseteq \mathbb{R}^{d_y}$. Then, if $g$ is convex-concave, it can be proved that $z^* \in \mathcal{Z}$ is a solution for (1) if and only if $z^* \in \mathcal{Z}$ is a solution for (3).

• **Fixed point problem.** Consider the fixed point problem:
$$\text{Find } z^* \in \mathbb{R}^d \text{ such that } T(z^*) = z^*, \tag{4}$$
where $T : \mathbb{R}^d \to \mathbb{R}^d$ is an operator. With $F(z) = z - T(z)$, it can be proved that $z^* \in \mathcal{Z} = \mathbb{R}^d$ is a solution for (1) if and only if $F(z^*) = 0$, i.e. $z^* \in \mathbb{R}^d$ is a solution for (4).

In order to prove convergence results, the following assumptions are introduced on the problem (1).

**Assumption 4.1** (Lipschitzness). Each operator $F_i$ is $L$-Lipschitz continuous on $\mathcal{Z}$, i.e. for all $u, v \in \mathcal{Z}$ we have $\|F_i(u) - F_i(v)\| \leq L\|u - v\|$.

For problems (2) and (3), $L$-Lipschitzness of the operator means that the functions $f(z)$ and $f(x, y)$ are $L$-smooth.

**Assumption 4.2** (Strong monotonicity). The operator $F$ is $\mu$-strongly monotone on $\mathcal{Z}$, i.e. for all $u, v \in \mathcal{Z}$ we have $\langle F(u) - F(v), u - v \rangle \geq \mu\|u - v\|^2$. Each operator $F_i$ is monotone on $\mathcal{Z}$, i.e. strongly monotone with $\mu = 0$.

For problems (2) and (3), strong monotonicity of $F$ means strong convexity of $f(z)$ and strong convexity-strong concavity of $f(x, y)$.

**Assumption 4.3** ($\delta$-relatedness in mean). The operators $\{F_i\}$ is $\delta$-related in mean on $\mathcal{Z}$. It means that for any $j$ operators $\{F_i - F_j\}$ is $\delta$-Lipschitz continuous in mean on $\mathcal{Z}$, i.e. for all $u, v \in \mathcal{Z}$ we have $\frac{1}{n}\sum_{i=1}^n \|F_i(u) - F_j(u) - F_i(v) + F_j(v)\|^2 \leq \delta^2 \|u - v\|^2$.

While Assumptions 4.1 and 4.2 are basic and widely known, Assumption 4.3 requires further remarks. This assumption goes back to the conditions of data similarity (as discussed in Section 2). In more detail, we consider distributed minimization (2) and saddle point (3) problems $f(z) = \frac{1}{n}\sum_{i=1}^n f_i(z), f(x, y) = \frac{1}{n}\sum_{i=1}^n f_i(x, y)$, and assume that for minimization local and global hessians are $\delta$-similar [79, 4, 90, 35, 51], i.e., $\|\nabla^2 f_j(z) - \nabla^2 f_i(z)\| \leq \delta$, and for SPP, second derivatives differ by $\delta$ [16, 51] $\|\nabla^2_{xx} f_j(x, y) - \nabla^2_{xx} f_i(x, y)\| \leq \delta, \|\nabla^2_{xy} f_j(x, y) - \nabla^2_{xy} f_i(x, y)\| \leq \delta, \|\nabla^2_{yy} f_j(x, y) - \nabla^2_{yy} f_i(x, y)\| \leq \delta$. In the worst case $\delta \sim L$, but often for practical machine learning problems $\delta < L$. In particular, if we assume that the data is uniformly distributed between devices, it can be proven (see Theorem 1.3 from [83], Theorem 2 from [35]) that $\delta = \tilde{\mathcal{O}}(L/\sqrt{b})$ or even $\delta = \tilde{\mathcal{O}}(L/b)$, where $b$ is the number of local data points on each of the devices.

Let us note some of the terms we use in the paper. By local steps/updates, we mean that some device $i$, using only local information about operator $F_i$, changes the local value of variable $z$. Local computations/calculations includes all computations of $F_i$ on a local device: local updates/steps may take place, or the value of the operator is accumulated and is sent to the other device without changes

---
**Algorithm 1** `Three Pillars Algorithm`
---
    **Parameters:** stepsizes $\gamma$ and $\eta$, momentum $\tau$, probability $p \in (0; 1]$, number of local steps $H$;
    **Initialization:** Choose $z^0 = m^0 = (x^0, y^0) \in \mathcal{Z}$;

1: **for** $k = 0, 1, \ldots, K - 1$ **do**
2:     Server takes $u_0^k = z^k$;
3:     **for** $t = 0, 1, \ldots, H - 1$ **do**
4:         Server computes $u_{t+1/2}^k = \mathrm{proj}_{\mathcal{Z}}[u_t^k - \eta(F_1(u_t^k) - F_1(m^k) + F(m^k) + \frac{1}{\gamma}(u_t^k - z^k - \tau(m^k - z^k)))]$;
5:         Server updates $u_{t+1}^k = \mathrm{proj}_{\mathcal{Z}}[u_t^k - \eta(F_1(u_{t+1/2}^k) - F_1(m^k) + F(m^k) + \frac{1}{\gamma}(u_{t+1/2}^k - z^k - \tau(m^k - z^k)))]$;
6:     **end for**
7:     Server broadcasts $u_H^k$ and $F_1(u_H^k)$ to devices;
8:     Devices in parallel compute $Q_i(F_i(m^k) - F_1(m^k) - F_i(u_H^k) + F_1(u_H^k))$ and send to server;
9:     Server updates $z^{k+1} = \mathrm{proj}_{\mathcal{Z}}[u_H^k + \gamma \cdot \frac{1}{n}\sum_{i=1}^{n} Q_i(F_i(m^k) - F_1(m^k) - F_i(u_H^k) + F_1(u_H^k))]$;
10:    Server updates $m^{k+1} = \begin{cases} z^k, & \text{with probability } p, \\ m^k, & \text{with probability } 1-p, \end{cases}$;
11:    **if** $m^{k+1} = z^k$ **then**
12:        Server broadcasts $m^{k+1}$ to devices;
13:        Devices in parallel compute $F_i(m^k)$ and send to server;
14:        Server computes $F(m^{k+1}) = \frac{1}{n}\sum_{i=1}^{n} F_i(m^{k+1})$;
15:    **end if**
16: **end for**
---

in local variables $z$. The local device $i$ complexity we mean the number of local computations of $F_i$ during the entire operation of algorithms, and by the total local complexity – the total number of computations of $F_i$ for all $i$. A communication round refers to establishing a connection between two devices and transmitting information in one or both directions. When multiple devices communicate in parallel—for instance, if all nodes send information to a central device—it is counted as a single communication round. Since we employ sparsifiers as compressors, we quantify the transmitted information by the number of coordinates sent. Communication complexity is the total volume of information exchanged between all devices during the execution of the algorithm. Communication time, on the other hand, is the duration spent on all such communication rounds.

## 5 Three pillars algorithm

### 5.1 The algorithm

We consider a centralized communication architecture where the main device is the server (node with number 1, which stores data associated with $F_1$). All other devices can send information to the server and receive responses from it. We follow the standard assumption that forwarding from devices to the server takes considerably longer than broadcasting from the server to the devices (see Section 3.5 from [43] and Section M.3 from [68]), hence we consider one-side compression from devices to the server. Our approach can be generalized to two-sided compression.

We split the idea of our new method `Three Pillars Algorithm` (Algorithm 1) into five parts.

• **Local problem.** The idea of the simplest local method is that the devices make a lot of independent gradient steps, using only their local operators/gradients and sometimes averaging the current value of the optimization variables [64, 66]. With this approach, between communications, the devices minimize their local functions and can move quite far away if the number of local steps is large [47]. Therefore, the estimates for `Local SGDA`, `FedAvg-S`, `ESTVGM` from Table 1 depend on the data heterogeneity parameter. This problem can be solved by modifying original local operators/functions via regularization. For example, we can ask to locally minimize not the local $f_i(z)$, but $f_i(z) + \lambda\|z - z_s\|^2$, where $x_s$ is the last point synchronized with all devices [45]. Statistical preconditioning, the popular trick for most methods under similarity conditions, is also a regularization of the local problem [79, 90]. It also can be considered as Mirror Descent with special Bregman divergence (see Section 1.1. of [35] for more details). The papers [51, 16] suggest looking at the stochastic preconditioning in distributed optimization from a different perspective. The authors propose to rewrite the original distributed problem as a composite problem of two terms: $\frac{1}{n}\sum_{i=1}^{n} f_i(x) = r(x) + q(x) = [f_1(x)] + [\frac{1}{n}\sum_{i=1}^{n} f_i(x) - f_1(x)]$. The proximal method works well for composite problems with proximal-friendly functions (e.g, simple $\ell_2$ regularizer) where the calculation of the proximal operator is for free, but, in the general case, the proximal operator is calculated by an auxiliary method with another iteration loop inside the main algorithm. The key idea in [51, 16]

is to use $r(x) = f_1(x)$ for the proximal operator. This implies the need of additional solving $\arg\min_x\{f_1(x) + \lambda\|x - x_s\|\}$, where $x_s$ is some point. Note that no commutation is needed to calculate the proximal operator as to solution needs access only to $f_1$ and can be solved locally. This reasoning shows that the idea with composite reformulation is also related to the regularization of the local problem. To extend the idea with composite reformulation into VI (see (2) for connections between min. prob. and VIs), we use the following proximal operator (see the loop in line 3 in Alg. 1):

$$\text{Find } \hat{u}^k \in \mathcal{Z} \text{ such that } \langle G(\hat{u}^k), z - \hat{u}^k \rangle \geq 0, \ \forall z \in \mathcal{Z}$$

with $G(z) = F_1(z) + \frac{1}{\gamma}(z - v^k)$ and $v^k = z^k + \tau(m^k - z^k) - \gamma \cdot (F(m^k) - F_1(m^k))$, where the essence of $m^k$ will be explained later. An important issue to be addressed is the selection of the basic method that should be used for an inexact calculation of the proximal operator (we will discuss it in the next paragraphs). We also need to underline one more point. Our approach solves the local problem (proximal operator) only on the server that owns $F_1$. It is possible to solve different local problems on all devices in parallel as proposed in [79], but this lead to non-optimal convergence guarantees. The method from [79] was revised in [90], which proposed local steps on one device and improved convergence rates. In [45], an algorithm was constructed that uses local updates on all devices and achieves the same convergence results as in [90] (only in the quadratic case). If no improvement is gained from running local updates on all devices, why should we do it?

• **Basic outer method.** We need a basic method for solving the composite VI. Gradient Descent is one option for minimization problems, but VI and SPP have their own optimal methods. Therefore, the basic outer method for Algorithm 1 also needs to be specialized for VI. In [51, 16], the autors used `Extra Gradient` [50, 71, 42]. In our work, we selected Tseng's method [84] for this task.

• **Local method.** Since the subproblem in the inner loop from line 3 in Algorithm 1 is a VI with a strongly monotone and Lipschitz continuous operator, then it can be solved using the `Extra Gradient` method [50]. It is optimal for this class of problems.

• **Use of compression.** In order to use compression in the method, an additional technique is also required. Following [75, 32, 15], we take the idea of basing on the variance reduction technique from [1]. We introduce a reference sequence $\{m^k\}_{k \geq 0}$ as in the classical variance reduction method [41]. At point $m^k$, we need to know the full values of operator $F$. When $m^k$ is updated (line 10), we transfer the full operators without compression (lines 12 and 13). If probability $p$ is small, $m^k$ is rarely modified and hence condition 11 is satisfied with low probability. It turns out that in Algorithm 1 at each iteration all nodes send a compressed package and very rarely transmit a full operator. If $p$ is chosen correctly, the effect of this is unnoticeable. But simply introducing an additional $m^k$ sequence is not enough, we also need to add a negative momentum: $\tau(m^k - z^k)$ (lines 4 and 5). This is dictated by the nature of the variance reduction method for VI.

• **Compression operators.** In Algorithm 1 (lines 8, 9) any unbiased compressor can be used, but to reveal the $\delta$-similarity ($\delta$-relatedness), we apply the so-called permutation compressors [82].

**Definition 5.1** (Permutation compressors [82]). For different cases of $n$ and $d$, we define
• **for $d \geq n$.** Assume that $d \geq n$ and $d = qn$, where $q \geq 1$ is an integer. Let $\pi = (\pi_1, \ldots, \pi_d)$ be a random permutation of $\{1, \ldots, d\}$. Then for all $u \in \mathbb{R}^d$ and each $i \in \{1, 2, \ldots, n\}$ we define $Q_i(u) = n \cdot \sum_{j=q(i-1)+1}^{qi} u_{\pi_j} e_{\pi_j}$.
• **for $d \leq n$.** Assume that $n \geq d$, $n > 1$ and $n = qd$, where $q \geq 1$ is an integer. Define the multiset $S := \{1, \ldots, 1, 2, \ldots, 2, \ldots, d, \ldots, d\}$, where each number occurs precisely $q$ times. Let $\pi = (\pi_1, \ldots, \pi_n)$ be a random permutation of $S$. Then for all $u \in \mathbb{R}^d$ and each $i \in \{1, 2, \ldots, n\}$ we define $Q_i(u) = du_{\pi_i} e_{\pi_i}$.

In practice, the case of $d \geq n$ is more relevant. Permutation compressors require generating a new permutation of numbers from $1$ to $d$ each iteration. To ensure the same permutations are generated on all devices, a random seed can be defined in each iteration in a predetermined way (e.g. iteration counter). The permutation organizes how compression operators correspond. Take $d = 12$ and $n = 3$: a permutation $(5, 2, 10, 7, 4, 12, 1, 9, 3, 8, 11, 6)$ directs the first device to send coordinates $5, 2, 10, 7$; the second $4, 12, 1, 9$; the third $3, 8, 11, 6$. This system ensures non-overlapping data transmission, reducing the data sent per device by a factor of $n$ and avoiding the zeroed "gradient" average that random selection risks. Permutation compressors thus correct redundancy and omission errors. They also capitalize on data similarity, ensuring the "gradient" at the server closely mirrors the true average. The original paper [82] recognized the benefits of this compressor with similar data but didn't fully leverage it due to its focus on non-convex settings without local steps.

• **Summary.** We use Tseng's method [84] for the composite VI, inaccurately computing the resolvent (proximal operator) with the `Extra Gradient` method [50]. Our convergence analysis is novel for Tseng's method and can be regarded as a byproduct and secondary contribution. We use a variance

reduction technique [1] for Tseng's method, remade into a compression technique in distributed settings [75, 32, 15], and choose a special compression for similarity setting [82].

## 5.2 Convergence

This section presents the convergence results of Algorithm 1 and its analysis.

**Theorem 5.2.** *Let $\{z^k\}_{k>0}$ denote the iterates of Algorithm 1 with compressors from Definition 5.1 for solving problem (1), which satisfies Assumptions 4.1, 4.2, 4.3. Then, if we choose the stepsizes $\gamma = \tilde{\mathcal{O}}(\min\{\frac{p}{\mu}, \frac{\sqrt{p}}{\delta}, \frac{H}{L}\})$, $\eta = \mathcal{O}((L + \frac{1}{\gamma})^{-1})$ and the momentum $\tau = p$ then we have the convergence guarantee $\mathbb{E}[\|z^K - z^*\|^2] \leq (1 - \frac{\gamma\mu}{2})^K \cdot 2\|z^0 - z^*\|^2$.*

In Algorithm 1, one mandatory communication round with compression occurs (lines 7 and 8) and possibly one more (without compression) with probability $p$ (lines 12 and 13). Permutation compressor compress by a factor of $n$; hence each iteration requires $\mathcal{O}\left(\frac{1}{n} + p\right)$ data transfers from devices to the server on average: in line 8, we use compression and forward $\frac{d}{n}$ coordinates from the devices to the server, in line 13, with probability $p$ we transfer the full vector of $d$ coordinates.

If $p$ is close to 1, Theorem 5.2 gives faster convergence, but more data transfer is needed. If $p$ tends to 0, the transmitted information complexity per iteration decreases but the iterative convergence rate drops. The optimal choice of $p$ is $\frac{1}{n}$, as stated in the corollary.

**Corollary 5.3.** *Under the conditions of Theorem 5.2, the following number of the outer iterations is needed to achieve the accuracy $\varepsilon$ (in terms of $\mathbb{E}\left[\|z - z^*\|^2\right] \lesssim \varepsilon$) by Algorithm 1 with $p = \frac{1}{n}$:*

$$\mathcal{O}([n + \frac{\delta\sqrt{n}}{\mu} + \frac{L}{\mu H}]\log\frac{\|z^0 - z^*\|^2}{\varepsilon}). \qquad (5)$$

Next, let us give a brief discussion of the previous estimate.

• **Communications and local calculations.** The iteration of Algorithm 1 has $(1 + p)$ communication rounds and forwards $\mathcal{O}\left(\frac{1}{n} + p\right)$ information on average. With $p = \frac{1}{n}$, the estimate in equation (5) is valid for both the number of iterations and communication rounds. To get an upper bound for the amount of information transmitted, divide (5) by $n$. The bound from Corollary 5.3 reflects the local complexity on all devices except the first (the server). To get an estimate for the first device, we need to multiply (5) by $H$.

• **Optimal choice of $H$.** It is clear from (5) that the number of local $H$ steps can improve the number of communications, but there is a limit beyond which more iterations are not useful. Taking $H = \lceil\frac{L}{\delta\sqrt{n}}\rceil$ is optimal. If there is no data similarity ($\delta \sim L$), then one can choose $H = 1$, and Algorithm 1 communication complexity estimate becomes similar to those of MASHA and Optimistic MASHA. Choosing $H = 1$ implies that local updates are meaningless when $\delta \sim L$. Papers [45, 88] on distributed minimization problems show that collecting large batches works just as well as methods that perform many local updates. Our paper considers deterministic methods, so we collect full batches and need local steps only within the presence of data similarity.

• **"Break" lower bounds.** $H$ from the previous paragraph yields estimates of $\tilde{\mathcal{O}}(n + \delta\sqrt{n}/\mu)$ and $\tilde{\mathcal{O}}(1 + \delta/\sqrt{n}\mu)$ for the number of communications and transmitted information, respectively. These results are better than any existing methods and even superior to lower bounds for deterministic methods. Note that our method uses randomized compression and hence lower bounds for deterministic methods does not apply for Algorithm 1 (see Table 1). To close this gap in Section 5.4 we provide new lower bound estimates that show the optimality of our approach.

• **Variable server.** In Algorithm 1, the server can be reselected deterministically or randomly at each outer iteration. Communication with the server can be decentralized using reduce and gather procedures [18], for this kind of procedures compression also greatly speeds up communication both theoretically and in practice [55]. Although the server can change, and decentralized communication networks can be used, it must still receive data from all devices. Asynchronous communication is not supported, the lack of communication with the device is also a problem for Algorithm 1. This is a potential problem that could slow down the method. This issue is partially solved by Algorithm 2 in the next section, which assumes communication with only 1 device per iteration.

## 5.3 Extension with partial participation

We also propose a modification of Algorithm 1 with partial participation (see Algorithm 2 in Appendix A). The essence of this method is to replace lines 8, 9 of Algorithm 1 with the following steps

---
8: Generate uniformly $i_k$ from $[n]$, device $i_k$ computes $F_{i_k}(u_H^k)$ and sends to server;
9: Server updates $z^{k+1} = \text{proj}_{\mathcal{Z}}[u_H^k + \gamma \cdot (F_{i_k}(m^k) - F_1(m^k) - F_{i_k}(u_H^k) + F_1(u_H^k))]$;

---

The following theorem is valid for Algorithm 2.

**Theorem 5.4.** *Let $\{z^k\}_{k>0}$ denote the iterates of Algorithm 2 for solving problem (1), which satisfies Assumptions 4.1, 4.2, 4.3. Then, if we choose the stepsizes $\gamma = \tilde{\mathcal{O}}(\min\{\frac{p}{\mu}, \frac{\sqrt{p}}{\delta}, \frac{H}{L}\})$, $\eta = \mathcal{O}((L + \frac{1}{\gamma})^{-1})$ and the momentum $\tau = p$ then we have the convergence $\mathbb{E}[\|z^K - z^*\|^2] \le (1 - \frac{\gamma\mu}{2})^K \cdot 2\|z^0 - z^*\|^2$.*

While in lines 8 and 9 of Algorithm 1 all devices compute the full local operators but forward only $\mathcal{O}(\frac{1}{n})$ information, in Algorithm 2 only one local device computes its operator and forwards this full operator. Hence, the total amount of transmitted information from all devices is equivalent for Algorithms 1 and 2. Therefore the optimal choice of $p$ for Algorithm 2 is the same as for Algorithm 1.

**Corollary 5.5.** *Under the conditions of Theorem 5.4, $\mathcal{O}([n + \frac{\delta\sqrt{n}}{\mu} + \frac{L}{\mu H}] \log \frac{\|z^0 - z^*\|^2}{\varepsilon})$ outer iterations are needed for the accuracy $\varepsilon$ (in terms of $\mathbb{E}[\|z - z^*\|^2] \lesssim \varepsilon$) by Algorithm 2 with $p = \frac{1}{n}$.*

Despite the fact that the estimates from Corollary 5.3 and 5.5 are the same. Algorithm 2 is better than Algorithm 1 in terms of local devices (but not the server) complexity. In line 8 of Algorithm 1 all devices collect full operator, while in Algorithm 2 only one node does it, then local complexity (not for first node) of Algorithm 2 is $n$ times better. But while Algorithm 1 and Algorithm 2 are the same in terms of the number of transmitted information, Algorithm 2 is $n$ times worse in terms of communication time. This is due to the fact that in line 8 of Algorithm 1, all devices in parallel send information using $\mathcal{O}(\frac{1}{n})$ units of time, while in Algorithm 2, one node need $\mathcal{O}(1)$ units of time.

## 5.4 Lower bounds

To derive lower bounds, we define a "full package" as the amount of information transmitted in one complete communication round from all devices to the server without compression. We start from the lower bounds for algorithms with partial participation, as in the previous section.

**Theorem 5.6.** *For any $\delta, \mu > 0$ and $n \in \mathbb{N}$ and $K \in \mathbb{N}$, there exists a distirbuted variational inequality (satisfying Assumptions 4.2 and 4.3) on $\mathbb{R}^d$ (where $d$ is sufficiently large) with $z^* \ne 0$, such that for any output $\hat{z}$ of any distributed algorithm with partial participation after sending $K$ full packages from the devices to the server, it holds that $\mathbb{E}[\|\hat{z} - z^*\|^2]$ is $\Omega(\exp(-16/(1 + \sqrt{\frac{\delta^2 n}{16\mu^2}} + 1) \cdot K)\|z^0 - z^*\|^2)$.*

The lower bounds confirm that results of the form $(1 + \delta/(\sqrt{n}\mu))$ are optimal for distributed methods with partial participation; thus, our algorithm achieves this optimality. Lower bounds serve to present a "worst-case" distributed VI where any algorithm with partial participation cannot outperform the established bounds. For a formal classification of distributed algorithms with partial participation, an exemplification of a problematic case, and proofs of the lower bounds, see Appendix C.3. For the methods with compression, we can also show that our result is optimal. But here we have to restrict the class of compression operators and consider only random sparsifiers – random choice of coordinates for forwarding (Permutation compressors are suitable). See Appendix C.4. How to obtain lower guarantees for arbitrary compressors remains an open question.

## 5.5 Extension with stochastic local computations

In this section, we consider a stochastic formulation of (1), namely, we assume that each local term from (1) also has the form of a finite sum: $F_i(z) = \frac{1}{M}\sum_{j=1}^{M} F_{i,j}(z)$. Although we can compute $F_i$ locally on each $i$th device, we would like to avoid this due to computational cost. The stochastic finite sum setting is common for machine learning and refers to Monte Carlo approach and empirical risk minimization [78]. Moreover, such a setup is more than natural especially in the context of similarity. For example, if we initially have a large dataset $N = nMb$ and distribute it to $n$ devices, then each device has $Mb$ samples: $F = \frac{1}{n}\sum_{i=1}^{n} F_i = \frac{1}{n}\sum_{i=1}^{n}[\frac{1}{M}\sum_{j=1}^{M} F_{i,j}] = \frac{1}{n}\sum_{i=1}^{n}[\frac{1}{M}\sum_{j=1}^{M}(\frac{1}{b}\sum_{l=1}^{b} F_{i,j,l})]$. The nature of similarity goes precisely that $F_i$ have the form of a sub-sum taken from a whole large sum. As we noted in Section 4, the larger the sample size $Mb$ for $F_i$, the smaller the similarity parameter $\delta$. Now in the stochastic case we consider $F_{i,j}$ and they also have similarity, but with a larger $\tilde{\delta}$, since $Mb$ is larger than $b$. To work with new setting we introduce two assumptions.

**Assumption 5.7** (Lipschitzness in mean). *Each operator $F_i$ is $\tilde{L}$-Lipschitz continuous in mean on $\mathcal{Z}$, i.e. for all $u, v \in \mathcal{Z}$ we have $\frac{1}{M}\sum_{j=1}^{M}\|F_{i,j}(u) - F_{i,j}(v)\|^2 \le \tilde{L}^2\|u - v\|^2$.*

**Assumption 5.8.** The operators $\{F_{i,j}\}$ is $\tilde{\delta}$-related in mean on $\mathcal{Z}$. For any $i_1$ operators $\{F_{i_1,j_1} - F_{i_2,j_2}\}$ is $\tilde{\delta}$-Lipschitz continuous in mean on $\mathcal{Z}$, i.e. for all $u, v \in \mathcal{Z}$ we have $\frac{1}{nM^2} \sum_{i_2=1}^{n} \sum_{j_1,j_2=1}^{M} \|F_{i_1,j_1}(u) - F_{i_2,j_2}(u) - F_{i_1,j_1}(v) + F_{i_2,j_2}(v)\|^2 \leq \tilde{\delta}^2 \|u-v\|^2$.

The modified version of Algorithm 1 for the new setting looks as follows (see the full version in Appendix A – Algorithm 3 ):

---

2: Server takes $u_0^k = w_0^k = z^k$;
4: Server computes
$\quad u_{t+1/2}^k = \text{proj}_{\mathcal{Z}}[\alpha u_t^k + (1-\alpha)w_t^k - \eta(F_1(w_t^k) - F_1(m^k) + F(m^k) + \frac{1}{\gamma}(w_t^k - z^k - \tau(m^k - z^k)))]$;
5: Server updates
$\quad u_{t+1}^k = \text{proj}_{\mathcal{Z}}[\alpha u_t^k + (1-\alpha)w_t^k - \eta(F_{1,j_t}(u_{t+1/2}^k) - F_{1,j_t}(w_t^k) + F_1(w_t^k) - F_1(m^k) + F(m^k)$
$\qquad + \frac{1}{\gamma}(u_{t+1/2}^k - z^k - \tau(m^k - z^k)))]$, where $j_t$ is generated uniformly from $[M]$;
$\quad$ Server updates $w_{t+1}^k = \begin{cases} u_{t+1}^k, & \text{with probability } q, \\ w_t^k, & \text{with probability } 1-q, \end{cases}$;
7: Server broadcasts $u_H^k$ and $F_{1,j_k^1}(u_H^k) - F_{1,j_k^1}(m^k)$ to devices, where $j_k^1$ is generated uniformly from $[M]$;
8: Devices in parallel compute $Q_i(F_{i,j_k^i}(m^k) - F_{1,j_k^1}(m^k) - F_{i,j_k^i}(u_H^k) + F_{1,j_k^1}(u_H^k))$ and send to server, where $j_k^i$ is generated uniformly from $[M]$;
9: Server updates
$\quad z^{k+1} = \text{proj}_{\mathcal{Z}}[u_H^k + \gamma \cdot \frac{1}{n}\sum_{i=1}^{n} Q_i(F_{i,j_k^i}(m^k) - F_{1,j_k^1}(m^k) - F_{i,j_k^i}(u_H^k) + F_{1,j_k^1}(u_H^k))]$;

---

To deal with stochasticity we make two modifications in Algorithm 1: in lines 2–5 we switch the local algorithm from classical deterministic `Extra Gradient` to stochastic `Extra Gradient` from [1] (Algorithm 1), in lines 7–9 we use stochastic operators for all $\{F_i\}$ instead of deterministic ones. The next theorem and corollary provides the convergence.

**Theorem 5.9.** *Let $\{z^k\}_{k\geq 0}$ denote the iterates of Algorithm 3 with compressors from Definition 5.1 for solving problem (1), which satisfies Assumptions 4.2, 5.7, 5.8. Then, if we choose the stepsizes $\gamma = \tilde{\mathcal{O}}(\min\{\frac{p}{\mu}, \frac{\sqrt{p}}{\tilde{\delta}}, \frac{H}{\sqrt{M}\tilde{L}}\})$, $\eta = \mathcal{O}(M^{-1/2}(\tilde{L} + \frac{1}{\gamma})^{-1})$, the momentums $\tau = p$, $\alpha = 1 - q$ and the probability $q = \frac{1}{M}$ then we have the convergence guarantee $\mathbb{E}[\|z^K - z^*\|^2] \leq (1 - \frac{\gamma\mu}{2})^K \cdot 2\|z^0 - z^*\|^2$.*

**Corollary 5.10.** *Under the conditions of Theorem 5.9 $\mathcal{O}([n + \frac{\tilde{\delta}\sqrt{n}}{\mu} + \frac{\sqrt{M}\tilde{L}}{\mu H}]\log \frac{\|z^0 - z^*\|^2}{\varepsilon})$ outer iterations are needed for the accuracy $\varepsilon$ (in terms of $\mathbb{E}[\|z - z^*\|^2] \lesssim \varepsilon$) by Alg.3 with $p = \frac{1}{n}$.*

### 5.6 Non-monotone case

In this section, we abandon the monotonicity assumption, but introduce the following assumption.

**Assumption 5.11** (Non-monotonicity). The operator $F$ is non-monotone (minty), if and only if there exists $z^* \in \mathbb{R}^d$ such that for all $z \in \mathbb{R}^d$ we have $\langle F(z), z - z^*\rangle \geq 0$.

This assumption is called the minty or variational stability condition. It is not a general non-monotonicity, but is already associated in the community with non-monotonicity [22, 38, 67, 61, 44, 37, 26], particularly with the setup, which is somewhat appropriate for GANS [59, 60, 27, 8].

**Theorem 5.12.** *Let $\{z^k\}_{k\geq 0}$ denote the iterates of Algorithm 1 for solving problem (1) with $\mathcal{Z} = \mathbb{R}^d$, which satisfies Assumptions 4.1, 5.11, 4.3. Then, if we additionally assume that $\|z^k\|, \|u_H^k\|, \|z^*\| \leq \Omega$ and choose the stepsizes $\gamma = \frac{\sqrt{p}}{6\delta}$, $\eta = \frac{1}{2(L+1/\gamma)}$, the momentum $\tau = p \leq \frac{1}{2}$ then $\frac{1}{K}\sum_{k=0}^{K-1}\mathbb{E}[\|F(u_H^k)\|^2] \lesssim \frac{\delta^2\Omega^2}{p}\left(\frac{1}{K} + \frac{1}{\sqrt{H}}\left(\frac{L\sqrt{p}}{\delta} + 1\right) + \frac{1}{H}\left(\frac{L\sqrt{p}}{\delta} + 1\right)^2\right)$.*

By substituting $H \sim K^2\left(\frac{L\sqrt{p}}{\delta} + 1\right)^2$, we can obtain that $\frac{1}{K}\sum_{k=0}^{K-1}\mathbb{E}[\|F(u_H^k)\|^2] \lesssim \frac{\delta^2\Omega^2}{pK}$. The discussion of the optimal choice of the parameter $p$, in this case, is the same as that following Theorem 5.2. Therefore, the following corollary is true.

**Corollary 5.13.** *Under the conditions of Theorem 5.12, $\mathcal{O}(\frac{n\delta^2\Omega^2}{\varepsilon^2})$ outer iterations is needed to achieve the accuracy $\varepsilon$ (in terms of $\mathbb{E}[\|F(z)\|^2] \lesssim \varepsilon^2$) by Algorithm 1 with $p = \frac{1}{n}$.*

One can note that the number of information transmitted is $n$ times less than the number of iterations in Corollary 5.13. Therefore, we have the communication complexity is equal to $\mathcal{O}(\delta^2\Omega^2/\varepsilon^2)$. This is better than the estimate $\mathcal{O}(L^2\Omega^2/\varepsilon^2)$ of a basic method that does not use local steps, similarity, and compression [33]. The problem is that our estimate of communication complexity does not

benefit from large $n$. In the strongly monotone case, however, we observed a more pleasant situation (see Table 1). Unfortunately, this is a standard problem of the non-monotone case. For example, the methods from [15] with compression have the same complexity $\mathcal{O}\left(L^2\Omega^2/\varepsilon^2\right)$ as the basic method without compression. The methods from the paper [11] converge only in the special case when all local operators are equal to each other. Other papers from Table 1 do not consider the non-monotone case or consider it under other assumptions, then their complexity results are hard to compare with ours. Finally, it turns out that our work is the first to get an acceleration in terms of communication complexity in the non-monotone case. On the negative side, we can note that the estimate on the number of local iterations $H$ in Theorem 5.12 is rather strict and grows with the number of iterations $K$. The results for Algorithm 2 in the non-monotone case can be obtained in a similar way.

## 6 Experiments

We conduct experiments on the robust linear regression problem. This problem is defined as $\min_{w\in\mathbb{R}^d}\max_{\|r_i\|\leq D}\frac{1}{2N}\sum_{i=1}^{N}(w^T(x_i+r_i)-y_i)^2+\frac{\lambda}{2}\|w\|^2-\frac{\beta}{2}\|r\|^2$, where $w$ are model weights, $\{x_i,y_i\}_{i=1}^N$ is the training dataset and $r$ is artificially added noise. We use $\ell_2$-regularization on both $w$ and $r$. We consider a network with $n=25$ devices and two types of datasets: synthetic and real. Synthetic data allows us to control the factor $\delta$, which measures statistical similarity of functions over different nodes. We take local datasets of size $b=100$ and generate the dataset $\{\hat{x}_i,\hat{y}_i\}_{i=1}^n$ at the server randomly, with each entry drawn from the Standard Gaussian distribution. The datasets at other devices ($i=2,\ldots,n$) are obtained by perturbing $\{\hat{x}_i,\hat{y}_i\}_{i=1}^n$ with normal noise $\xi_i$, having zero mean and controlled variance. Real datasets are taken from LibSVM library [19]. We uniformly divide the whole dataset between devices. The solution $z^*$ is found using a large number of iterations of `Extra Gradient` [50, 42] method. Algorithms 1 and 2 are compared with methods from Table 1 (`Extra Gradient` [50, 42], `Local SGDA` [25], `FedAvg-S` [36], `SCAFFOLD-S` [36], `SCAFFOLD-Catalyst-S` [36], `ESTVGM` [11], `SMMDS` [16], `MASHA` [15], `Optimistic MASHA` [12], `Accelerated Extra Gradient` [51]) implemented in Python 3.7. All the methods are tuned as in the theory of the corresponding papers. The number of full operator transmissions by one of the arbitrary devices is used as the $x$-asis. The experiment results are shown in Figure 1. It is evident that Algorithm 1 and 2 consistently outperform all other competing algorithms by a significant margin. Furthermore, as the value of $\delta$ decreases, the superiority over `Local SGDA`, `SCAFFOLD-S`, `SCAFFOLD-Catalyst-S`, `MASHA`, and `Optimistic MASHA` becomes even more pronounced.

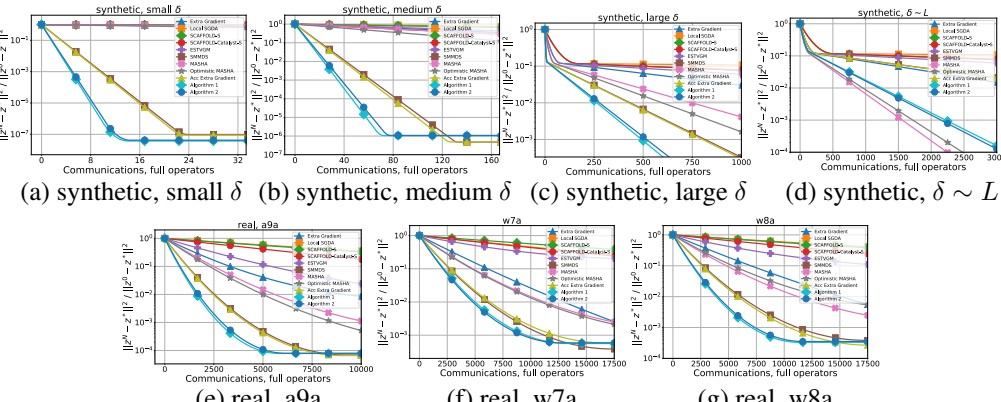

(a) synthetic, small $\delta$    (b) synthetic, medium $\delta$    (c) synthetic, large $\delta$    (d) synthetic, $\delta\sim L$

(e) real, a9a        (f) real, w7a        (g) real, w8a

Figure 1: Comparison of state-of-the-art methods for distributed VI. The comparison is made on synthetic datasets with small, medium, and large $\delta$, as well as the real datasets a9a, w7a, w8a from LibSVM. The $x$-axis denotes the number of full operators transmitted by one of the devices.

## 7 Conclusion and future works

We presented two new algorithms for solving distributed VI and SPP: Algorithm 1, which combines compression, similarity, and local steps techniques; and Algorithm 2, which uses partial participation instead of compression. Both algorithms have the best communication complexity, outperforming existing methods in theory and practice. For future work we plan to i) derive new methods to stochastic settings (finite-sum or bounded variance setups); ii) obtain lower bounds on communication complexity with compression.

## Acknowledgments

This research of A. Beznosikov has been supported by The Analytical Center for the Government of the Russian Federation (Agreement No. 70-2021-00143 dd. 01.11.2021, IGK 000000D730321P5Q0002).

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
