5:         Server updates

$$u_{t+1}^k = \text{proj}_{\mathcal{Z}} \left[ u_t^k - \eta \left( F_1(u_{t+1/2}^k) - F_1(m^k) + F(m^k) + \tfrac{1}{\gamma}(u_{t+1/2}^k - z^k - \tau(m^k - z^k)) \right) \right];$$

6:     **end for**
7:     Server broadcasts $u_H^k$ to devices;
8:     Generate uniformly $i_k$ from $[n]$;
9:     Device $i_k$ computes $F_{i_k}(u_H^k)$ and sends to server;
10:    Server updates $z^{k+1} = \text{proj}_{\mathcal{Z}} \left[ u_H^k + \gamma \cdot (F_{i_k}(m^k) - F_1(m^k) - F_{i_k}(u_H^k) + F_1(u_H^k)) \right]$;
11:    Server updates $m^{k+1} = \begin{cases} z^k, & \text{with probability } p, \\ m^k, & \text{with probability } 1 - p, \end{cases}$;
12:    **if** $m^{k+1} = z^k$ **then**
13:        Server broadcasts $m^{k+1}$ to devices;
14:        All devices in parallel compute $F_i(m^k)$ and send to server;
15:        Server takes all $F_i(m^k)$, saves them and computes $F(m^{k+1}) = \tfrac{1}{n} \sum_{i=1}^{n} F_i(m^{k+1})$;
16:    **end if**
17: **end for**

---

---

**Algorithm 3** `Three Pillars Algorithm with Variance Reduction`

---

**Parameters:** stepsizes $\gamma$ and $\eta$, momentums $\tau$ and $\alpha$, probabilities $p$ and $q \in (0;1]$ , number of local steps $H$;

**Initialization:** Choose $z^0 = m^0 = (x^0, y^0) \in \mathcal{Z}$;

1: **for** $k = 0, 1, \ldots, K - 1$ **do**
2: $\quad$ Server takes $u_0^k = w_0^k = z^0$;
3: $\quad$ **for** $t = 0, 1, \ldots, H - 1$ **do**
4: $\qquad$ Server computes
$$u_{t+1/2}^k = \text{proj}_{\mathcal{Z}}[\alpha u_t^k + (1 - \alpha)w_t^k - \eta(F_1(w_t^k) - F_1(m^k) + F(m^k)$$
$$+ \tfrac{1}{\gamma}(w_t^k - z^k - \tau(m^k - z^k)))];$$
5: $\qquad$ Server updates
$$u_{t+1}^k = \text{proj}_{\mathcal{Z}}[\alpha u_t^k + (1 - \alpha)w_t^k - \eta(F_{1,j_t}(u_{t+1/2}^k) - F_{1,j_t}(w_t^k) + F_1(w_t^k) - F_1(m^k) + F(m^k)$$
$$+ \tfrac{1}{\gamma}(u_{t+1/2}^k - z^k - \tau(m^k - z^k)))], \text{ where } j_t \text{ is generated uniformly from } [M];$$
6: $\qquad$ Server updates $w_{t+1}^k = \begin{cases} u_{t+1}^k, & \text{with probability } q, \\ w_t^k, & \text{with probability } 1 - q, \end{cases};$
7: $\quad$ **end for**
8: $\quad$ Server broadcasts $u_H^k$ and $F_{1,j_k^1}(u_H^k) - F_{1,j_k^1}(m^k)$ to devices, where $j_k^1$ is generated uniformly from $[M]$;
9: $\quad$ Devices in parallel compute $Q_i(F_{i,j_k^i}(m^k) - F_{1,j_k^1}(m^k) - F_{i,j_k^i}(u_H^k) + F_{1,j_k^1}(u_H^k))$ and send to server, where $j_k^i$ is generated uniformly from $[M]$;
10: $\quad$ Server updates
$$z^{k+1} = \text{proj}_{\mathcal{Z}}[u_H^k + \gamma \cdot \tfrac{1}{n}\textstyle\sum_{i=1}^{n} Q_i(F_{i,j_k^i}(m^k) - F_{1,j_k^1}(m^k) - F_{i,j_k^i}(u_H^k) + F_{1,j_k^1}(u_H^k))];$$
11: $\quad$ Server updates $m^{k+1} = \begin{cases} z^k, & \text{with probability } p, \\ m^k, & \text{with probability } 1 - p, \end{cases};$
12: $\quad$ **if** $m^{k+1} = z^k$ **then**
13: $\qquad$ Server broadcasts $m^{k+1}$ to devices;
14: $\qquad$ All devices in parallel compute $F_i(m^k)$ and send to server;
15: $\qquad$ Server takes all $F_i(m^k)$, saves them and computes $F(m^{k+1}) = \tfrac{1}{n}\sum_{i=1}^{n} F_i(m^{k+1})$;
16: $\quad$ **end if**
17: **end for**

---

## B Auxiliary facts

**Lemma B.1** (see Theorems 1 and 2 from [82])**.** *The Permutation compressors from Definition 5.1 are unbiased and satisfy*

$$\mathbb{E}\left[\left\|\frac{1}{n}\sum_{i=1}^{n}Q_i(a_i) - \frac{1}{n}\sum_{i=1}^{n}a_i\right\|^2\right] \le \frac{1}{n}\sum_{i=1}^{n}\left\|a_i - \frac{1}{n}\sum_{j=1}^{n}a_j\right\|^2$$

*for all $a_1, \ldots, a_n \in \mathbb{R}^d$.*

## C Missing proofs

### C.1 Proof of Theorem 5.2

The loop in line 3 runs $H$ steps of the Extra Gradient method for the problem:

$$\text{Find } \hat{u}^k \in \mathcal{Z} \text{ such that } \langle G(\hat{u}^k), z - \hat{u}^k \rangle \ge 0, \ \forall z \in \mathcal{Z}$$

$$\text{with } G(z) = F_1(z) - F_1(m^k) + F(m^k) + \frac{1}{\gamma}(z - z^k - \tau(m^k - z^k)) \tag{6}$$

The notation for the solution $\hat{u}^k$ of (6) will be used in the proofs. We start the proof with the following lemma.

**Lemma C.1.** *Let $\{z^k\}_{k\ge 0}$ be the sequence generated by Algorithm 1 for solving problem* (1)*, which satisfies Assumptions 4.1,4.2, 4.3. Then we have the following estimate on one iteration:*

$$\mathbb{E}\left[\left\|z^{k+1} - z^*\right\|^2\right] + \mathbb{E}\left[\|m^{k+1} - z^*\|^2\right]$$

$$\le \left(1 - \tau + p - \frac{\gamma\mu}{2}\right)\mathbb{E}\left[\left\|z^k - z^*\right\|^2\right] + \left(1 + \tau - p - \frac{\gamma\mu}{2}\right)\mathbb{E}\left[\|m^k - z^*\|^2\right]$$

$$+ \left(2 + \frac{4\gamma\delta^2}{\mu} + \frac{4}{\gamma\mu} + 8\gamma^2\delta^2\right)\mathbb{E}\left[\|u^k - \hat{u}^k\|^2\right]$$

$$- \left(1 - \tau - \frac{3\gamma\mu}{2}\right)\mathbb{E}\left[\left\|z^k - \hat{u}^k\right\|^2\right] - \left(\tau - \frac{3\gamma\mu}{2} - 8\gamma^2\delta^2\right)\mathbb{E}\left[\|m^k - \hat{u}^k\|^2\right].$$

**Proof:** Let us define an additional notation $w^k = u_H^k + \gamma \cdot \frac{1}{n}\sum_{i=1}^{n}Q_i(F_i(m^k) - F_1(m^k) - F_i(u_H^k) + F_1(u_H^k))$. With this notation, one can note that $z^{k+1} = \text{proj}_{\mathcal{Z}}[w^k]$ (line 9). Using the non-expansiveness of the Euclidean projection and making small rearrangements, we have

$$\mathbb{E}\left[\left\|z^{k+1} - z^*\right\|^2\right]$$

$$= \mathbb{E}\left[\left\|\text{proj}_{\mathcal{Z}}\left[w^k\right] - \text{proj}_{\mathcal{Z}}\left[z^*\right]\right\|^2\right]$$

$$\le \mathbb{E}\left[\left\|w^k - z^*\right\|^2\right]$$

$$= \mathbb{E}\left[\left\|z^k - z^*\right\|^2\right] + 2\mathbb{E}\left[\langle w^k - z^k, z^k - z^*\rangle\right] + \mathbb{E}\left[\left\|w^k - z^k\right\|^2\right]$$

$$= \mathbb{E}\left[\left\|z^k - z^*\right\|^2\right] + 2\mathbb{E}\left[\langle w^k - z^k, \hat{u}^k - z^*\rangle\right] + 2\mathbb{E}\left[\langle w^k - z^k, z^k - \hat{u}^k\rangle\right] + \mathbb{E}\left[\left\|w^k - z^k\right\|^2\right]$$

$$= \mathbb{E}\left[\left\|z^k - z^*\right\|^2\right] + 2\mathbb{E}\left[\langle w^k - z^k, \hat{u}^k - z^*\rangle\right] + \mathbb{E}\left[\left\|w^k - \hat{u}^k\right\|^2\right] - \mathbb{E}\left[\left\|z^k - \hat{u}^k\right\|^2\right]$$

$$= \mathbb{E}\left[\left\|z^k - z^*\right\|^2\right]$$

$$+ 2\mathbb{E}\left[\langle u_H^k + \gamma \cdot \frac{1}{n}\sum_{i=1}^{n}Q_i(F_i(m^k) - F_1(m^k) - F_i(u_H^k) + F_1(u_H^k)) - z^k, \hat{u}^k - z^*\rangle\right]$$

$$+ \mathbb{E}\left[\left\|w^k - \hat{u}^k\right\|^2\right] - \mathbb{E}\left[\left\|z^k - \hat{u}^k\right\|^2\right]$$

$$= \mathbb{E}\left[\left\|z^k - z^*\right\|^2\right]$$

$$+ 2\mathbb{E}\left[\left\langle u_H^k + \gamma \cdot \mathbb{E}_Q\left[\frac{1}{n}\sum_{i=1}^n Q_i(F_i(m^k) - F_1(m^k) - F_i(u_H^k) + F_1(u_H^k))\right] - z^k, \hat{u}^k - z^*\right\rangle\right]$$

$$+ \mathbb{E}\left[\left\|w^k - \hat{u}^k\right\|^2\right] - \mathbb{E}\left[\left\|z^k - \hat{u}^k\right\|^2\right]$$

$$= \mathbb{E}\left[\left\|z^k - z^*\right\|^2\right] + 2\mathbb{E}\left[\langle u_H^k + \gamma \cdot (F(m^k) - F_1(m^k)) - z^k, \hat{u}^k - z^*\rangle\right]$$

$$- 2\gamma\mathbb{E}\left[\langle F(u_H^k) - F_1(u_H^k), \hat{u}^k - z^*\rangle\right] + \mathbb{E}\left[\left\|w^k - \hat{u}^k\right\|^2\right] - \mathbb{E}\left[\left\|z^k - \hat{u}^k\right\|^2\right].$$

In the last step, we also use the unbiasedness (Lemma B.1) of permutation compressors and their independence from previous points and iterations (in particular, from $z^k, u_H^k, \hat{u}^k$). With one more auxiliary notation $v^k = z^k + \tau(m^k - z^k) - \gamma \cdot (F(m^k) - F_1(m^k))$, we have

$$\mathbb{E}\left[\left\|z^{k+1} - z^*\right\|^2\right] \leq \mathbb{E}\left[\left\|z^k - z^*\right\|^2\right] + 2\mathbb{E}\left[\langle u_H^k - v^k, \hat{u}^k - z^*\rangle\right] + \tau\mathbb{E}\left[\langle m^k - z^k, \hat{u}^k - z^*\rangle\right]$$

$$- 2\gamma\mathbb{E}\left[\langle F(u_H^k) - F_1(u_H^k), \hat{u}^k - z^*\rangle\right] + \mathbb{E}\left[\left\|w^k - \hat{u}^k\right\|^2\right] - \mathbb{E}\left[\left\|z^k - \hat{u}^k\right\|^2\right]$$

$$= \mathbb{E}\left[\left\|z^k - z^*\right\|^2\right] + 2\mathbb{E}\left[\langle \hat{u}^k - v^k, \hat{u}^k - z^*\rangle\right] + \tau\mathbb{E}\left[\langle m^k - z^k, \hat{u}^k - z^*\rangle\right]$$

$$- 2\gamma\mathbb{E}\left[\langle F(u_H^k) - F_1(u_H^k), \hat{u}^k - z^*\rangle\right] + 2\mathbb{E}\left[\langle u_H^k - \hat{u}^k, \hat{u}^k - z^*\rangle\right]$$

$$+ \mathbb{E}\left[\left\|w^k - \hat{u}^k\right\|^2\right] - \mathbb{E}\left[\left\|z^k - \hat{u}^k\right\|^2\right].$$

Invoking the optimality of $\hat{u}^k$ (the solution of the subproblem (6)): $\langle \gamma F_1(\hat{u}^k) + \hat{u}^k - v^k, \hat{u}^k - z\rangle \leq 0$ (for all $z \in \mathcal{Z}$), yields:

$$\mathbb{E}\left[\left\|z^{k+1} - z^*\right\|^2\right]$$

$$\leq \mathbb{E}\left[\left\|z^k - z^*\right\|^2\right] - 2\gamma\mathbb{E}\left[\langle F_1(\hat{u}^k), \hat{u}^k - z^*\rangle\right] - 2\gamma\mathbb{E}\left[\langle F(u_H^k) - F_1(u_H^k), \hat{u}^k - z^*\rangle\right]$$

$$+ \tau\mathbb{E}\left[\langle m^k - z^k, \hat{u}^k - z^*\rangle\right] + 2\mathbb{E}\left[\langle u_H^k - \hat{u}^k, \hat{u}^k - z^*\rangle\right] + \mathbb{E}\left[\left\|w^k - \hat{u}^k\right\|^2\right] - \mathbb{E}\left[\left\|z^k - \hat{u}^k\right\|^2\right]$$

$$= \mathbb{E}\left[\left\|z^k - z^*\right\|^2\right] - 2\gamma\mathbb{E}\left[\langle F_1(\hat{u}^k), \hat{u}^k - z^*\rangle\right] - 2\gamma\mathbb{E}\left[\langle F(\hat{u}^k) - F_1(\hat{u}^k), \hat{u}^k - z^*\rangle\right]$$

$$+ 2\mathbb{E}\left[\langle \gamma(F(\hat{u}^k) - F_1(\hat{u}^k) - F(u_H^k) + F_1(u_H^k)) + u_H^k - \hat{u}^k, \hat{u}^k - z^*\rangle\right]$$

$$+ \tau\mathbb{E}\left[\langle m^k - z^k, \hat{u}^k - z^*\rangle\right] + \mathbb{E}\left[\left\|w^k - \hat{u}^k\right\|^2\right] - \mathbb{E}\left[\left\|z^k - \hat{u}^k\right\|^2\right].$$

Using the optimality of the solution $z^*$: $\langle \gamma F(z^*), z^* - z\rangle \leq 0$ (for all $z \in \mathcal{Z}$ and for $\hat{u}^k$, in particular) along with the $\mu$-strong monotonicity of $F$ (Assumption 4.2), we obtain

$$\mathbb{E}\left[\left\|z^{k+1} - z^*\right\|^2\right] \leq \mathbb{E}\left[\left\|z^k - z^*\right\|^2\right] - 2\gamma\mathbb{E}\left[\langle F(\hat{u}^k) - F(z^*), \hat{u}^k - z^*\rangle\right]$$

$$+ 2\mathbb{E}\left[\langle \gamma(F(\hat{u}^k) - F_1(\hat{u}^k) - F(u_H^k) + F_1(u_H^k)) + u_H^k - \hat{u}^k, \hat{u}^k - z^*\rangle\right]$$

$$+ \tau\mathbb{E}\left[\langle m^k - z^k, \hat{u}^k - z^*\rangle\right] + \mathbb{E}\left[\left\|w^k - \hat{u}^k\right\|^2\right] - \mathbb{E}\left[\left\|z^k - \hat{u}^k\right\|^2\right]$$

$$\leq \mathbb{E}\left[\left\|z^k - z^*\right\|^2\right] - 2\gamma\mu\mathbb{E}\left[\left\|\hat{u}^k - z^*\right\|^2\right]$$

$$+ 2\mathbb{E}\left[\langle \gamma(F(\hat{u}^k) - F_1(\hat{u}^k) - F(u_H^k) + F_1(u_H^k)) + u_H^k - \hat{u}^k, \hat{u}^k - z^*\rangle\right]$$

$$+ \tau\mathbb{E}\left[\langle m^k - z^k, \hat{u}^k - z^*\rangle\right] + \mathbb{E}\left[\left\|w^k - \hat{u}^k\right\|^2\right] - \mathbb{E}\left[\left\|z^k - \hat{u}^k\right\|^2\right].$$

The equality $2\langle a, b\rangle = \|a + b\|^2 - \|a\|^2 - \|b\|^2$ gives

$$\mathbb{E}\left[\left\|z^{k+1} - z^*\right\|^2\right] \leq \mathbb{E}\left[\left\|z^k - z^*\right\|^2\right] - 2\gamma\mu\mathbb{E}\left[\left\|\hat{u}^k - z^*\right\|^2\right]$$

$$+ 2\mathbb{E}\left[\langle \gamma(F(\hat{u}^k) - F_1(\hat{u}^k) - F(u_H^k) + F_1(u_H^k)) + u_H^k - \hat{u}^k, \hat{u}^k - z^*\rangle\right]$$

$$+ \tau\mathbb{E}\left[\langle m^k - \hat{u}^k, \hat{u}^k - z^*\rangle\right] + \tau\mathbb{E}\left[\langle \hat{u}^k - z^k, \hat{u}^k - z^*\rangle\right]$$

$$+ \mathbb{E}\left[\left\|w^k - \hat{u}^k\right\|^2\right] - \mathbb{E}\left[\left\|z^k - \hat{u}^k\right\|^2\right]$$

$$= \mathbb{E}\left[\left\|z^k - z^*\right\|^2\right] - 2\gamma\mu\mathbb{E}\left[\left\|\hat{u}^k - z^*\right\|^2\right]$$

$$+ 2\mathbb{E}\left[\langle \gamma(F(\hat{u}^k) - F_1(\hat{u}^k) - F(u_H^k) + F_1(u_H^k)) + u_H^k - \hat{u}^k, \hat{u}^k - z^*\rangle\right]$$

$$+ \tau \mathbb{E}\left[\|m^k - z^*\|^2\right] - \tau \mathbb{E}\left[\|m^k - \hat{u}^k\|^2\right] - \tau \|\hat{u}^k - z^*\|^2$$
$$+ \tau \mathbb{E}\left[\|\hat{u}^k - z^k\|^2\right] + \tau \mathbb{E}\left[\|\hat{u}^k - z^*\|^2\right] - \tau \mathbb{E}\left[\|z^k - z^*\|^2\right]$$
$$+ \mathbb{E}\left[\|w^k - \hat{u}^k\|^2\right] - \mathbb{E}\left[\|z^k - \hat{u}^k\|^2\right]$$
$$= (1-\tau)\mathbb{E}\left[\left\|z^k - z^*\right\|^2\right] + \tau \mathbb{E}\left[\|m^k - z^*\|^2\right] - 2\gamma\mu\mathbb{E}\left[\left\|\hat{u}^k - z^*\right\|^2\right]$$
$$+ 2\mathbb{E}\left[\langle \gamma(F(\hat{u}^k) - F_1(\hat{u}^k) - F(u_H^k) + F_1(u_H^k)) + u_H^k - \hat{u}^k, \hat{u}^k - z^*\rangle\right]$$
$$+ \mathbb{E}\left[\left\|w^k - \hat{u}^k\right\|^2\right] - (1-\tau)\mathbb{E}\left[\left\|z^k - \hat{u}^k\right\|^2\right] - \tau\mathbb{E}\left[\|m^k - \hat{u}^k\|^2\right].$$

By Young's inequalities $2\langle a, b\rangle \leq \|a\|^2 + \|b\|^2$ and $\|a+b\|^2 \leq 2\|a\|^2 + 2\|b\|^2$, we have

$$\mathbb{E}\left[\left\|z^{k+1} - z^*\right\|^2\right]$$
$$\leq (1-\tau)\mathbb{E}\left[\left\|z^k - z^*\right\|^2\right] + \tau\mathbb{E}\left[\|m^k - z^*\|^2\right] - 2\gamma\mu\mathbb{E}\left[\left\|\hat{u}^k - z^*\right\|^2\right]$$
$$+ \frac{2}{\gamma\mu}\mathbb{E}\left[\left\|\gamma(F(\hat{u}^k) - F_1(\hat{u}^k) - F(u_H^k) + F_1(u_H^k)) + u_H^k - \hat{u}^k\right\|^2\right] + \frac{\gamma\mu}{2}\mathbb{E}\left[\left\|\hat{u}^k - z^*\right\|^2\right]$$
$$+ \mathbb{E}\left[\left\|w^k - \hat{u}^k\right\|^2\right] - (1-\tau)\mathbb{E}\left[\left\|z^k - \hat{u}^k\right\|^2\right] - \tau\mathbb{E}\left[\|m^k - \hat{u}^k\|^2\right]$$
$$= (1-\tau)\mathbb{E}\left[\left\|z^k - z^*\right\|^2\right] + \tau\mathbb{E}\left[\|m^k - z^*\|^2\right] - \frac{3\gamma\mu}{2}\mathbb{E}\left[\left\|\hat{u}^k - z^*\right\|^2\right]$$
$$+ \frac{4\gamma}{\mu}\mathbb{E}\left[\left\|F(\hat{u}^k) - F_1(\hat{u}^k) - F(u_H^k) + F_1(u_H^k)\right\|^2\right] + \frac{4}{\gamma\mu}\mathbb{E}\left[\left\|u_H^k - \hat{u}^k\right\|^2\right]$$
$$+ \mathbb{E}\left[\left\|u_H^k + \gamma \cdot \frac{1}{n}\sum_{i=1}^n Q_i(F_i(m^k) - F_1(m^k) - F_i(u_H^k) + F_1(u_H^k)) - \hat{u}^k\right\|^2\right]$$
$$- (1-\tau)\mathbb{E}\left[\left\|z^k - \hat{u}^k\right\|^2\right] - \tau\mathbb{E}\left[\|m^k - \hat{u}^k\|^2\right]$$
$$= (1-\tau)\mathbb{E}\left[\left\|z^k - z^*\right\|^2\right] + \tau\mathbb{E}\left[\|m^k - z^*\|^2\right] - \frac{3\gamma\mu}{2}\mathbb{E}\left[\left\|\hat{u}^k - z^*\right\|^2\right]$$
$$+ \frac{4\gamma}{\mu}\mathbb{E}\left[\left\|F(\hat{u}^k) - F_1(\hat{u}^k) - F(u_H^k) + F_1(u_H^k)\right\|^2\right] + \frac{4}{\gamma\mu}\mathbb{E}\left[\left\|u_H^k - \hat{u}^k\right\|^2\right]$$
$$+ 2\mathbb{E}\left[\left\|u_H^k - \hat{u}^k\right\|^2\right] + 2\gamma^2\mathbb{E}\left[\left\|\frac{1}{n}\sum_{i=1}^n Q_i(F_i(m^k) - F_1(m^k) - F_i(u_H^k) + F_1(u_H^k))\right\|^2\right]$$
$$- (1-\tau)\mathbb{E}\left[\left\|z^k - \hat{u}^k\right\|^2\right] - \tau\mathbb{E}\left[\|m^k - \hat{u}^k\|^2\right]. \tag{7}$$

With the equality $\|a\|^2 = \|a - b\|^2 + 2\langle a, b\rangle - \|b\|^2$, definition of $Q_i$ and Lemma B.1, we obtain

$$\mathbb{E}\left[\left\|\frac{1}{n}\sum_{i=1}^n Q_i(F_i(m^k) - F_1(m^k) - F_i(u_H^k) + F_1(u_H^k))\right\|^2\right]$$
$$= \mathbb{E}\left[\left\|\frac{1}{n}\sum_{i=1}^n Q_i(F_i(m^k) - F_1(m^k) - F_i(u_H^k) + F_1(u_H^k)) - (F(m^k) - F_1(m^k) - F(u_H^k) + F_1(u_H^k))\right\|^2\right]$$
$$+ 2\mathbb{E}\left[\left\langle \frac{1}{n}\sum_{i=1}^n Q_i(F_i(m^k) - F_1(m^k) - F_i(u_H^k) + F_1(u_H^k)), F(m^k) - F_1(m^k) - F(u_H^k) + F_1(u_H^k)\right\rangle\right]$$
$$- \mathbb{E}\left[\left\|F(m^k) - F_1(m^k) - F(u_H^k) + F_1(u_H^k)\right\|^2\right]$$
$$= \mathbb{E}\left[\left\|\frac{1}{n}\sum_{i=1}^n Q_i(F_i(m^k) - F_1(m^k) - F_i(u_H^k) + F_1(u_H^k)) - (F(m^k) - F_1(m^k) - F(u_H^k) + F_1(u_H^k))\right\|^2\right]$$

$$+ 2\mathbb{E}\left[\left\langle \frac{1}{n}\sum_{i=1}^{n}\mathbb{E}_{Q_i}\left[Q_i(F_i(m^k) - F_1(m^k) - F_i(u_H^k) + F_1(u_H^k))\right], F(m^k) - F_1(m^k) - F(u_H^k) + F_1(u_H^k)\right\rangle\right]$$

$$- \mathbb{E}\left[\left\|F(m^k) - F_1(m^k) - F(u_H^k) + F_1(u_H^k)\right\|^2\right]$$

$$=\mathbb{E}\left[\left\|\frac{1}{n}\sum_{i=1}^{n}Q_i(F_i(m^k) - F_1(m^k) - F_i(u_H^k) + F_1(u_H^k)) - (F(m^k) - F_1(m^k) - F(u_H^k) + F_1(u_H^k))\right\|^2\right]$$

$$+ 2\mathbb{E}\left[\langle F(m^k) - F_1(m^k) - F(u_H^k) + F_1(u_H^k), F(m^k) - F_1(m^k) - F(u_H^k) + F_1(u_H^k)\rangle\right]$$

$$- \mathbb{E}\left[\left\|F(m^k) - F_1(m^k) - F(u_H^k) + F_1(u_H^k)\right\|^2\right]$$

$$=\mathbb{E}\left[\mathbb{E}_{Q_i}\left[\left\|\frac{1}{n}\sum_{i=1}^{n}Q_i(F_i(m^k) - F_1(m^k) - F_i(u_H^k) + F_1(u_H^k)) - (F(m^k) - F_1(m^k) - F(u_H^k) + F_1(u_H^k))\right\|^2\right]\right]$$

$$+ \mathbb{E}\left[\left\|F(m^k) - F_1(m^k) - F(u_H^k) + F_1(u_H^k)\right\|^2\right]$$

$$\leq\frac{1}{n}\sum_{i=1}^{n}\mathbb{E}\left[\left\|F_i(m^k) - F_1(m^k) - F_i(u_H^k) + F_1(u_H^k) - (F(m^k) - F_1(m^k) - F_m(u_H^k) + F_1(u_H^k))\right\|^2\right]$$

$$+ \mathbb{E}\left[\left\|F(m^k) - F_1(m^k) - F(u_H^k) + F_1(u_H^k)\right\|^2\right]$$

$$=\frac{1}{n}\sum_{i=1}^{n}\mathbb{E}\left[\left\|F_i(m^k) - F_1(m^k) - F_i(u_H^k) + F_1(u_H^k)\right\|^2\right]$$

$$- \frac{2}{n}\sum_{i=1}^{n}\mathbb{E}\left[\langle F_i(m^k) - F_1(m^k) - F_i(u_H^k) + F_1(u_H^k), F(m^k) - F_1(m^k) - F_m(u_H^k) + F_1(u_H^k)\rangle\right]$$

$$+ \frac{1}{n}\sum_{i=1}^{n}\mathbb{E}\left[\left\|F(m^k) - F_1(m^k) - F(u_H^k) + F_1(u_H^k)\right\|^2\right]$$

$$+ \mathbb{E}\left[\left\|F(m^k) - F_1(m^k) - F(u_H^k) + F_1(u_H^k)\right\|^2\right]$$

$$=\frac{1}{n}\sum_{i=1}^{n}\mathbb{E}\left[\left\|F_i(m^k) - F_1(m^k) - F_i(u_H^k) + F_1(u_H^k)\right\|^2\right].$$

Hence, using Assumption 4.3, we can deduce

$$\mathbb{E}\left[\left\|\frac{1}{n}\sum_{i=1}^{n}Q_i(F(m^k) - F_1(m^k) - F_i(u_H^k) + F_1(u_H^k))\right\|^2\right] \leq \delta^2 \left\|m^k - u_H^k\right\|^2 \tag{8}$$

$$\left\|F(\hat{u}^k) - F_1(\hat{u}^k) - F(u_H^k) + F_1(u_H^k)\right\|^2$$

$$= \left\|\frac{1}{n}\sum_{i=1}^{n}\left(F_i(\hat{u}^k) - F_1(\hat{u}^k) - F_i(u_H^k) + F_1(u_H^k)\right)\right\|^2$$

$$\leq \frac{1}{n}\sum_{i=1}^{n}\left\|F_i(\hat{u}^k) - F_1(\hat{u}^k) - F_i(u_H^k) + F_1(u_H^k)\right\|^2 \leq \delta^2 \left\|\hat{u}^k - u_H^k\right\|^2. \tag{9}$$

One can substitute (8), (9) in (7) and get

$$\mathbb{E}\left[\left\|z^{k+1} - z^*\right\|^2\right]$$

$$\leq (1 - \tau)\mathbb{E}\left[\left\|z^k - z^*\right\|^2\right] + \tau\mathbb{E}\left[\left\|m^k - z^*\right\|^2\right] - \frac{3\gamma\mu}{2}\mathbb{E}\left[\left\|\hat{u}^k - z^*\right\|^2\right]$$

$$+ \frac{4\gamma\delta^2}{\mu}\mathbb{E}\left[\left\|u_H^k - \hat{u}^k\right\|^2\right] + \frac{4}{\gamma\mu}\mathbb{E}\left[\left\|u_H^k - \hat{u}^k\right\|^2\right]$$

$$+ 2\mathbb{E}\left[\left\|u_H^k - \hat{u}^k\right\|^2\right] + 2\gamma^2\delta^2\mathbb{E}\left[\left\|m^k - u_H^k\right\|^2\right]$$

$$- (1-\tau)\mathbb{E}\left[\|z^k - \hat{u}^k\|^2\right] - \tau\mathbb{E}\left[\|m^k - \hat{u}^k\|^2\right]$$

$$\leq (1-\tau)\mathbb{E}\left[\|z^k - z^*\|^2\right] + \tau\mathbb{E}\left[\|m^k - z^*\|^2\right] - \frac{3\gamma\mu}{2}\mathbb{E}\left[\|\hat{u}^k - z^*\|^2\right]$$

$$+ \left(2 + \frac{4\gamma\delta^2}{\mu} + \frac{4}{\gamma\mu} + 4\gamma^2\delta^2\right)\mathbb{E}\left[\|u_H^k - \hat{u}^k\|^2\right]$$

$$- (1-\tau)\mathbb{E}\left[\|z^k - \hat{u}^k\|^2\right] - (\tau - 4\gamma^2\delta^2)\mathbb{E}\left[\|m^k - \hat{u}^k\|^2\right]$$

$$= (1-\tau)\mathbb{E}\left[\|z^k - z^*\|^2\right] + \tau\mathbb{E}\left[\|m^k - z^*\|^2\right] - \frac{3\gamma\mu}{4}\mathbb{E}\left[\|\hat{u}^k - z^*\|^2\right] - \frac{3\gamma\mu}{4}\mathbb{E}\left[\|\hat{u}^k - z^*\|^2\right]$$

$$+ \left(2 + \frac{4\gamma\delta^2}{\mu} + \frac{4}{\gamma\mu} + 4\gamma^2\delta^2\right)\mathbb{E}\left[\|u_H^k - \hat{u}^k\|^2\right]$$

$$- (1-\tau)\mathbb{E}\left[\|z^k - \hat{u}^k\|^2\right] - (\tau - 4\gamma^2\delta^2)\mathbb{E}\left[\|m^k - \hat{u}^k\|^2\right]. \tag{10}$$

Here we again use Young's inequality $\|a + b\|^2 \leq 2\|a\|^2 + 2\|b\|^2$. The update of $m^{k+1}$ (line 10) gives

$$\mathbb{E}_{m^{k+1}}\left[\|m^{k+1} - z^*\|^2\right] = (1-p)\|m^k - z^*\|^2 + p\|z^k - z^*\|^2, \tag{11}$$

Then, summing up (10) and (11), we have

$$\mathbb{E}\left[\|z^{k+1} - z^*\|^2 + \|m^{k+1} - z^*\|^2\right] \leq (1 - \tau + p)\mathbb{E}\left[\|z^k - z^*\|^2\right] + (1 + \tau - p)\mathbb{E}\left[\|m^k - z^*\|^2\right]$$

$$- \frac{3\gamma\mu}{4}\mathbb{E}\left[\|\hat{u}^k - z^*\|^2\right] - \frac{3\gamma\mu}{4}\mathbb{E}\left[\|\hat{u}^k - z^*\|^2\right]$$

$$+ \left(2 + \frac{4\gamma\delta^2}{\mu} + \frac{4}{\gamma\mu} + 4\gamma^2\delta^2\right)\mathbb{E}\left[\|u_H^k - \hat{u}^k\|^2\right]$$

$$- (1-\tau)\mathbb{E}\left[\|z^k - \hat{u}^k\|^2\right] - (\tau - 4\gamma^2\delta^2)\mathbb{E}\left[\|m^k - \hat{u}^k\|^2\right].$$

Finally, using $\|a + b\|^2 \geq \frac{2}{3}\|a\|^2 - 2\|b\|^2$, we obtain

$$\mathbb{E}\left[\|z^{k+1} - z^*\|^2\right] + \mathbb{E}\left[\|m^{k+1} - z^*\|^2\right]$$

$$\leq \left(1 - \tau + p - \frac{\gamma\mu}{2}\right)\mathbb{E}\left[\|z^k - z^*\|^2\right] + \left(1 + \tau - p - \frac{\gamma\mu}{2}\right)\mathbb{E}\left[\|m^k - z^*\|^2\right]$$

$$+ \left(2 + \frac{4\gamma\delta^2}{\mu} + \frac{4}{\gamma\mu} + 4\gamma^2\delta^2\right)\mathbb{E}\left[\|u_H^k - \hat{u}^k\|^2\right]$$

$$- \left(1 - \tau - \frac{3\gamma\mu}{2}\right)\mathbb{E}\left[\|z^k - \hat{u}^k\|^2\right] - \left(\tau - \frac{3\gamma\mu}{2} - 4\gamma^2\delta^2\right)\mathbb{E}\left[\|m^k - \hat{u}^k\|^2\right].$$

$$\square$$

Now we are ready to prove the main theorem.

**Theorem C.2** (Theorem 5.2). *Let $\{z^k\}_{k \geq 0}$ denote the iterates of Algorithm 1 for solving problem (1), which satisfies Assumptions 4.1, 4.2, 4.3. Then, if we choose the stepsizes $\gamma$, $\eta$ and the momentum $\tau$ as follows:*

$$\tau = p \leq \frac{1}{4}, \quad \gamma = \min\left\{\frac{p}{3\mu}, \frac{\sqrt{p}}{4\delta}, \frac{1}{L} \cdot \left(\frac{H}{4\log\frac{40L}{\mu p}} - 1\right)\right\}, \quad \eta = \frac{1}{4\left(L + \frac{1}{\gamma}\right)}, \quad H \geq 8\log\frac{40L}{\mu p}, \tag{12}$$

*we have the convergence:*

$$\mathbb{E}\left[\|z^K - z^*\|^2\right] \leq \left(1 - \frac{\gamma\mu}{2}\right)^K \cdot 2\|z^0 - z^*\|^2.$$

**Proof:** The loop in line 3 makes $H$ steps of the Extra Gradient method for the problem (6). Note that the operator in (6) is $\left(L + \frac{1}{\gamma}\right)$-Lipschitz continuous (Assumption 4.1) and $\frac{1}{\gamma}$-strongly monotone (Assumption 4.2). For this case, there are convergence guarantees in the literature (see Theorem 1

from [30]):

$$\text{with} \quad \eta = \frac{1}{4\left(L + \frac{1}{\gamma}\right)} \quad \text{it holds} \quad \left\|u_H^k - \hat{u}^k\right\|^2 \leq \exp\left(-\frac{\frac{1}{\gamma} \cdot H}{4\left(L + \frac{1}{\gamma}\right)}\right) \left\|z^k - \hat{u}^k\right\|^2$$

$$= \exp\left(-\frac{H}{4\left(\gamma L + 1\right)}\right) \left\|z^k - \hat{u}^k\right\|^2.$$

If $\gamma \leq \frac{1}{L} \cdot \left(\frac{H}{4\log\frac{40L}{\mu p}} - 1\right)$ and $H \geq 8\log\frac{40L}{\mu p}$, then one can obtain

$$\left\|u_H^k - \hat{u}^k\right\|^2 \leq \frac{\mu p}{40L} \left\|z^k - \hat{u}^k\right\|^2.$$

Combining this fact and Lemma C.1 gives

$$\mathbb{E}\left[\left\|z^{k+1} - z^*\right\|^2\right] + \mathbb{E}\left[\|m^{k+1} - z^*\|^2\right]$$

$$\leq \left(1 - \tau + p - \frac{\gamma\mu}{2}\right) \mathbb{E}\left[\left\|z^k - z^*\right\|^2\right] + \left(1 + \tau - p - \frac{\gamma\mu}{2}\right) \mathbb{E}\left[\|m^k - z^*\|^2\right]$$

$$+ \left(2 + \frac{4\gamma\delta^2}{\mu} + \frac{4}{\gamma\mu} + 4\gamma^2\delta^2\right) \frac{\mu p}{40L} \left\|z^k - \hat{u}^k\right\|^2$$

$$- \left(1 - \tau - \frac{3\gamma\mu}{2}\right) \left\|z^k - \hat{u}^k\right\|^2 - \left(\tau - \frac{3\gamma\mu}{2} - 4\gamma^2\delta^2\right) \|m^k - \hat{u}^k\|^2.$$

With the choice of $\gamma$, $H$ and $p$ from (12), we get that

$$2 + \frac{4\gamma\delta^2}{\mu} + \frac{4}{\gamma\mu} + 4\gamma^2\delta^2 \leq 2 + \frac{4\delta^2}{\mu} \cdot \frac{\sqrt{p}}{4\delta} + \frac{4}{\mu} \cdot \max\left\{\frac{3\mu}{p}, \frac{4\delta}{\sqrt{p}}, L\left(\frac{H}{4\log\frac{40L}{\mu p}} - 1\right)^{-1}\right\} + 4\delta^2 \cdot \left(\frac{\sqrt{p}}{4\delta}\right)^2$$

$$\leq 2 + \frac{\delta\sqrt{p}}{\mu} + \frac{4}{\mu} \cdot \max\left\{\frac{3\mu}{p}, \frac{4\delta}{\sqrt{p}}, L\right\} + \frac{p}{2}$$

$$\leq 3 + 17\max\left\{\frac{1}{p}, \frac{\delta}{\mu\sqrt{p}}, \frac{L}{\mu}\right\}$$

$$\leq \frac{20L}{\mu p}.$$

Then, we have

$$\mathbb{E}\left[\left\|z^{k+1} - z^*\right\|^2\right] + \mathbb{E}\left[\|m^{k+1} - z^*\|^2\right]$$

$$\leq \left(1 - \tau + p - \frac{\gamma\mu}{2}\right) \mathbb{E}\left[\left\|z^k - z^*\right\|^2\right] + \left(1 + \tau - p - \frac{\gamma\mu}{2}\right) \mathbb{E}\left[\|m^k - z^*\|^2\right]$$

$$+ \frac{1}{2}\left\|z^k - \hat{u}^k\right\|^2$$

$$- \left(1 - \tau - \frac{3\gamma\mu}{2}\right) \left\|z^k - \hat{u}^k\right\|^2 - \left(\tau - \frac{3\gamma\mu}{2} - 4\gamma^2\delta^2\right) \|m^k - \hat{u}^k\|^2$$

$$\leq \left(1 - \tau + p - \frac{\gamma\mu}{2}\right) \mathbb{E}\left[\left\|z^k - z^*\right\|^2\right] + \left(1 + \tau - p - \frac{\gamma\mu}{2}\right) \mathbb{E}\left[\|m^k - z^*\|^2\right]$$

$$- \left(\frac{1}{2} - \tau - \frac{3\gamma\mu}{2}\right) \left\|z^k - \hat{u}^k\right\|^2 - \left(\tau - \frac{3\gamma\mu}{2} - 4\gamma^2\delta^2\right) \|m^k - \hat{u}^k\|^2.$$

With the choice $\tau$ from (12), we obtain

$$\mathbb{E}\left[\left\|z^{k+1} - z^*\right\|^2\right] + \mathbb{E}\left[\|m^{k+1} - z^*\|^2\right]$$

$$\leq \left(1 - \frac{\gamma\mu}{2}\right) \mathbb{E}\left[\left\|z^k - z^*\right\|^2\right] + \left(1 - \frac{\gamma\mu}{2}\right) \mathbb{E}\left[\|m^k - z^*\|^2\right]$$

$$- \left(\frac{1}{4} - \frac{3\gamma\mu}{2}\right) \left\|z^k - \hat{u}^k\right\|^2 - \left(p - \frac{3\gamma\mu}{2} - 4\gamma^2\delta^2\right) \|m^k - \hat{u}^k\|^2.$$

The proof is close to be completed by choosing $\gamma$ according to (12):

$$\mathbb{E}\left[\left\|z^{k+1} - z^*\right\|^2 + \|m^{k+1} - z^*\|^2\right] \leq \left(1 - \frac{\gamma\mu}{2}\right) \mathbb{E}\left[\left\|z^k - z^*\right\|^2 + \|m^k - z^*\|^2\right].$$

It remains to run the recursion and to take into account that $m^0 = z^0$.

$\square$

## C.2 Proof of Theorem 5.4

**Lemma C.3.** *Let $\{z^k\}_{k\geq 0}$ be the sequence generated by Algorithm 2 for solving problem* (1), *which satisfies Assumptions 4.1,4.2, 4.3. Then we have the following estimate on one iteration:*

$$\mathbb{E}\left[\left\|z^{k+1} - z^*\right\|^2\right] + \mathbb{E}\left[\left\|m^{k+1} - z^*\right\|^2\right]$$

$$\leq \left(1 - \tau + p - \frac{\gamma\mu}{2}\right)\mathbb{E}\left[\left\|z^k - z^*\right\|^2\right] + \left(1 + \tau - p - \frac{\gamma\mu}{2}\right)\mathbb{E}\left[\left\|m^k - z^*\right\|^2\right]$$

$$+ \left(2 + \frac{4\gamma\delta^2}{\mu} + \frac{4}{\gamma\mu} + 4\gamma^2\delta^2\right)\mathbb{E}\left[\left\|u^k - \hat{u}^k\right\|^2\right]$$

$$- \left(1 - \tau - \frac{3\gamma\mu}{2}\right)\mathbb{E}\left[\left\|z^k - \hat{u}^k\right\|^2\right] - \left(\tau - \frac{3\gamma\mu}{2} - 4\gamma^2\delta^2\right)\mathbb{E}\left[\left\|m^k - \hat{u}^k\right\|^2\right].$$

**Proof:** The proof is almost exactly the same as that of Lemma C.1. It is sufficient to replace $\left[\frac{1}{n}\sum_{i=1}^n Q_i(F(m^k) - F_1(m^k) - F_i(u_H^k) + F_1(u_H^k))\right]$ in Lemma C.1 with $\left[F_{i_k}(m^k) - F_1(m^k) - F_{i_k}(u^k) + F_1(u^k)\right]$. In Lemma C.1, we need to estimate $\mathbb{E}_Q\left[\frac{1}{n}\sum_{i=1}^n Q_i(F_i(m^k) - F_1(m^k) - F_i(u_H^k) + F_1(u_H^k))\right]$, $\mathbb{E}_{Q_i}\left[Q_i(F_i(m^k) - F_1(m^k) - F_i(u_H^k) + F_1(u_H^k))\right]$ and $\left\|\frac{1}{n}\sum_{i=1}^n Q_i(F(m^k) - F_1(m^k) - F_i(u_H^k) + F_1(u_H^k))\right\|$, now one need to use the following bounds:

$$\mathbb{E}_{i_k}\left[F_{i_k}(m^k) - F_1(m^k) - F_{i_k}(u^k) + F_1(u^k)\right] = F(m^k) - F_1(m^k) - F(u^k) + F_1(u^k)$$

$$\mathbb{E}_{i_k}\left[\left\|F_{i^k}(m^k) - F_1(m^k) - F_{i^k}(u^k) + F_1(u^k)\right\|^2\right]$$

$$= \frac{1}{n}\sum_{i=1}^n \left\|F_i(m^k) - F_1(m^k) - F_i(u^k) + F_1(u^k)\right\|^2 \leq \delta^2 \left\|m^k - u^k\right\|^2.$$

Here we use that $i_k$ is generated uniformly and independently. This estimate is the same with (8), then (10) is valid for Algorithm 2. Therefore, the rest of the proof is also correct.

$\square$

**Theorem C.4** (Theorem 5.4). *Let $\{z^k\}_{k\geq 0}$ denote the iterates of Algorithm 2 for solving problem* (1), *which satisfies Assumptions 4.1, 4.2, 4.3. Then, if we choose the stepsizes $\gamma$, $\eta$ and the momentum $\tau$ as follows:*

$$\tau = p \leq \frac{1}{4}, \quad \gamma = \min\left\{\frac{p}{3\mu}, \frac{\sqrt{p}}{4\delta}, \frac{1}{L} \cdot \left(\frac{H}{4\log\frac{40L}{\mu p}} - 1\right)\right\}, \quad \eta = \frac{1}{4\left(L + \frac{1}{\gamma}\right)}, \quad H \geq 8\log\frac{40L}{\mu p},$$

*we have the convergence:*

$$\mathbb{E}\left[\left\|z^K - z^*\right\|^2\right] \leq \left(1 - \frac{\gamma\mu}{2}\right)^K \cdot 2\left\|z^0 - z^*\right\|^2.$$

**Proof:** The proof is identical to that of Theorem 5.2 (Theorem C.2).

$\square$

## C.3 Proof of Theorem 5.6

One of the basic techniques for obtaining lower bounds is to construct some "bad" example of the problem, on which any algorithm performs no better than the lower estimate claims. As we mentioned in the main part, variational inequalities are a broad class of problems, in particular, they include min-max (saddle point problems). We will consider this problem to derive the lower bounds. The paper [16] gives lower bounds and a "bad" example for distributed deterministic saddle point problems. We will modify their approach. In more details, we look at the distributed saddle point problem:

$$\min_{x\in\mathbb{R}^d} \max_{y\in\mathbb{R}^d} f(x,y) = \frac{1}{n}\sum_{i=1}^n f_i(x,y). \tag{13}$$

In Section 4, we highlight the operator $F(z) = F(x,y) = [\nabla_x f(x,y), -\nabla_y f(x,y)]$. Next, we rewrite Assumptions 4.2 and 4.3 for (13).

**Assumption C.5.** The function $f$ is $\mu$-strongly convex – strongly concave on on $\mathbb{R}^d \times \mathbb{R}^d$, i.e. for all $x_1, x_2 \in \mathbb{R}^d$, $y_1, y_2 \in \mathbb{R}^d$ we have

$$\langle \nabla_x f(x_1, y_1) - \nabla_x f(x_2, y_2); x_1 - x_2 \rangle - \langle \nabla_y f(x_1, y_1) - \nabla_y f(x_2, y_2); y_1 - y_2 \rangle$$
$$\geq \mu \left( \|x_1 - x_2\|^2 + \|y_1 - y_2\|^2 \right).$$

Each function $f_i$ is convex–concave on $\mathbb{R}^d \times \mathbb{R}^d$, i.e. strongly convex – strongly concave with $\mu = 0$.

**Assumption C.6.** The functions $\{f_i\}$ is $\delta$-related in mean on $\mathbb{R}^d \times \mathbb{R}^d$, i.e. for any $j$ and all $x_1, x_2 \in \mathbb{R}^d$, $y_1, y_2 \in \mathbb{R}^d$ we have

$$\frac{1}{n} \sum_{i=1}^{n} \left( \|\nabla_x f_i(x_1, y_1) - \nabla_x f_j(x_1, y_1) - \nabla_x f_i(x_2, y_2) + \nabla_x f_j(x_2, y_2)\|^2 \right.$$

$$\left. + \|\nabla_y f_i(x_1, y_1) - \nabla_y f_j(x_1, y_1) - \nabla_y f_i(x_2, y_2) + \nabla_y f_j(x_2, y_2)\|^2 \right)$$
$$\leq \delta^2 \left( \|x_1 - x_2\|^2 + \|y_1 - y_2\|^2 \right).$$

Next, we need to formally define the class of algorithms for which we will obtain lower bounds.

**Definition C.7** (Class of algorithms). Each device $i$ has its own local memories $\mathcal{M}_i^x$ and $\mathcal{M}_i^y$ for the $x$- and $y$-variables –with initialization $\mathcal{M}_m^x = \mathcal{M}_m^y = \{0\}$. $\mathcal{M}_1^x$ and $\mathcal{M}_1^y$ are memories of the server. These memories $\mathcal{M}_m^x$ and $\mathcal{M}_m^x$ can be updated as follows:

• **Communication from devices to server:** During one communication round we sample uniformly and independently batch $S$ of any size $b$ from $(1 + [n-1])$ and ask devices with number from $S$ to share some vector of their local memories with the server, i.e. can add points $x, y$ to the local memories of the server according to the next rule:

$$x \in \text{span}\left\{ x_1, \bigcup_{i \in S} x_i \right\}, \quad y \in \text{span}\left\{ y_1, \bigcup_{i \in S} y_i \right\},$$

where $x_j \in \mathcal{M}_j^x$ and $y_j \in \mathcal{M}_j^y$. If the batch size is equal to $b$ we say that transmits $b$ full packages from devices to the server (e.g., by sending the full vector with dimension $d$).Batch of the size $(n-1)$ is equal to the situation, when all devices send their memories to the server.

• **Communication from server to devices:** Since in the results for upper bounds we fight for the communications from the devices to the server, we assume that broadcasting from the server to the devices is free and for all $i$ we can add any number of points $x, y$ to the local memories:

$$x \in \text{span}\left\{ x_1, x_i \right\}, \quad y \in \text{span}\left\{ y_1, y_i \right\},$$

where $x_j \in \mathcal{M}_j^x$ and $y_j \in \mathcal{M}_j^y$.

• **Local computations:** During local computations each device $i$ can make any computations using $f_i$, i.e. can add points $x, y$ to the local memories:

$$x \in \text{span}\{x' , \nabla_x f_i(x'', y'')\}, \quad y \in \text{span}\{y' , \nabla_y f_i(x'', y'')\},$$

for given $x', x'' \in \mathcal{M}_i^x$ and $y', y'' \in \mathcal{M}_i^y$.

The final global output is calculated as $\hat{x} \in \mathcal{M}_1^x$, $\hat{y} \in \mathcal{M}_1^y$.

Since here we consider partial participation regime, we assume that during one round the number of communicating devices can be random, but if a device communicates with the server, it communicates fully (without compression) by sending the whole vector.

Now we are ready to construct the "bad" problem. We use the following bilinearly function:

$$f(x, y) = \frac{\delta}{4\sqrt{n}} x^T A y + \frac{\mu}{2} \|x\|^2 - \frac{\mu}{2} \|y\|^2 + \frac{\delta^2}{16 n \mu} e_1^T y, \tag{14}$$

where $e_j$ is the $j$th basis vector ($j$th coordinate is equal to 1) and

$$A = \begin{pmatrix} 1 & -1 & & & & & & \\ & 1 & -1 & & & & & \\ & & 1 & -1 & & & & \\ & & & 1 & -1 & & & \\ & & & & \cdots & \cdots & & \\ & & & & & 1 & -1 & \\ & & & & & & 1 & -1 \\ & & & & & & & 1 \end{pmatrix}$$

Then we construct functions $f_i$. Let us define a vector $a_q$ as the $q$th row of the matrix $A$. Using this notation we split function $f$ to finite sum $\frac{1}{n}\sum_{i=1}^{n} f_i$ in following way:

$$f_1(x, y) = \frac{\mu}{2}\|x\|^2 - \frac{\mu}{2}\|y\|^2 + \frac{\delta^2}{16\mu}e_1^T y$$

$$f_i(x, y) = \frac{\delta\sqrt{n}}{4}x^T\left[\sum_{j\equiv(i-1)\bmod(n-1)} e_j a_j^T\right] y + \frac{\mu}{2}\|x\|^2 - \frac{\mu}{2}\|y\|^2 \quad \text{for} \quad i > 2. \tag{15}$$

We define $A_i$ as $A_i = \left[\sum_{j\equiv(i-1)\bmod(n-1)} e_j a_j^T\right]$ and $A_1 = 0$.

**Lemma C.8.** *The problem* (14) *with $f_i$ from* (15) *satisfies Assumptions C.5 and C.6 with $\mu$, $\delta$.*

**Proof:** It is easy to verify that $f$ is $\mu$ - strongly-convex-strongly-concave. We need to check that $\{f_i\}$ are $\delta$-related in mean:

$$\frac{1}{n}\sum_{i=1}^{n}\left[\|\nabla_x f_i(x_1, y_1) - \nabla_x f_j(x_1, y_1) - \nabla_x f_i(x_2, y_2) + \nabla_x f_j(x_2, y_2)\|^2\right.$$

$$\left. + \|\nabla_y f_i(x_1, y_1) - \nabla_y f_j(x_1, y_1) - \nabla_y f_i(x_2, y_2) + \nabla_y f_j(x_2, y_2)\|^2\right]$$

$$= \frac{1}{n}\sum_{i=1}^{n}\left[\left\|\sqrt{n}\cdot\frac{\delta}{4}(A_i - A_j)(y_1 - y_2)\right\|^2 + \left\|\sqrt{n}\cdot\frac{\delta}{4}(A_i - A_j)^T(x_2 - x_1)\right\|^2\right]$$

$$= \frac{\delta^2}{8}\sum_{i=1}^{n}\left[\|A_i(y_1 - y_2)\|^2 + \|A_j(y_1 - y_2)\|^2 + \|A_i^T(x_1 - x_2)\|^2 + \|A_j^T(x_1 - x_2)\|^2\right]$$

$$= \frac{\delta^2}{8}\sum_{i=1}^{n}\left[(y_1 - y_2)^T A_i^T A_i \|y_1 - y_2\|^2 + (x_1 - x_2)^T A_i A_i^T(x_1 - x_2)\right]$$

$$+ \frac{\delta^2}{8}\cdot\lambda_{\max}(A_j^T A_j)\left[\|y_1 - y_2\|^2 + \|x_1 - x_2\|^2\right]$$

$$= \frac{\delta^2}{8}\sum_{i=2}^{n}(y_1 - y_2)^T\left[\sum_{l\equiv(i-1)\bmod(n-1)} a_l e_l^T\right]\left[\sum_{l\equiv(i-1)\bmod(n-1)} e_l a_l^T\right](y_1 - y_2)$$

$$+ \frac{\delta^2}{8}\sum_{i=2}^{n}(x_1 - x_2)^T\left[\sum_{l\equiv(i-1)\bmod(n-1)} e_l a_l^T\right]\left[\sum_{l\equiv(i-1)\bmod(n-1)} a_l e_l^T\right](x_1 - x_2)$$

$$+ \frac{\delta^2}{8}\cdot\lambda_{\max}(A_j^T A_j)\left[\|y_1 - y_2\|^2 + \|x_1 - x_2\|^2\right]$$

$$= \frac{\delta^2}{8}\sum_{i=2}^{n}(y_1 - y_2)^T\left[\sum_{l\equiv(i-1)\bmod(n-1)} a_l a_l^T\right](y_1 - y_2)$$

$$+ \frac{\delta^2}{8}\sum_{i=2}^{n}(x_1 - x_2)^T\left[\sum_{l\equiv(i-1)\bmod(n-1)} a_l^T a_l e_l e_l^T\right](x_1 - x_2)$$

$$+ \frac{\delta^2}{8}\cdot\lambda_{\max}(A_j^T A_j)\left[\|y_1 - y_2\|^2 + \|x_1 - x_2\|^2\right]$$

$$= \frac{\delta^2}{8}(y_1 - y_2)^T\left[\sum_{l=1}^{d} a_l a_l^T\right](y_1 - y_2) + \frac{\delta^2}{4}(x_1 - x_2)^T\left[\sum_{l=1}^{d} e_l e_l^T\right](x_1 - x_2)$$

$$+ \frac{\delta^2}{8}\cdot\lambda_{\max}(A_j^T A_j)\left[\|y_1 - y_2\|^2 + \|x_1 - x_2\|^2\right]$$

$$\leq \frac{\delta^2}{8}\cdot\lambda_{\max}\left(\sum_{l=1}^{d} a_l a_l^T\right)\|y_1 - y_2\|^2 + \frac{\delta^2}{4}\|x_1 - x_2\|^2$$

$$+ \frac{\delta^2}{8} \cdot \lambda_{\max}(A_j^T A_j) \left[ \|y_1 - y_2\|^2 + \|x_1 - x_2\|^2 \right].$$

It remains to note that $\lambda_{\max}\left( \sum_{l=1}^{d} a_l a_l^T \right) \leq 4$ and $\lambda_{\max}(A_j^T A_j) \leq 4$.

$\square$

Next, we need to understand how fast we come close to the solution of our problem (14) + (15) depending on the number of communication rounds.

**Lemma C.9.** *Let the problem* (14) + (15) *be solved by any method that satisfies Definition C.7. Then after sending $K$ full packages from the devices to the server, in expectation only the first $K$ coordinates of the final output can be non-zero while the rest of the $d - K$ coordinates are strictly equal to zero.*

**Proof:** The initial global output is simply the zero vectors since $\mathcal{M}_1^x = \{0\}$ and $\mathcal{M}_1^y = \{0\}$. Our goal is to understand how many non-zero coordinates we will have after sending $K$ full packages from the devices to the server. We begin with introducing some notation: let

$$E_0 := \{0\}, \quad E_K := \mathrm{span}\{e_1, \ldots, e_K\}.$$

The initialization gives $\mathcal{M}_i^x = E_0$, $\mathcal{M}_i^y = E_0$.

Let us analyze how $\mathcal{M}_i^x, \mathcal{M}_i^y$ can change by performing only local computations. For the $i$th device we add to $\mathcal{M}_i^x, \mathcal{M}_i^y$ the following points:

$$x \in \mathrm{span}\{x', A_i y'\}, \quad y \in \mathrm{span}\{e_1 \cdot \mathbb{I}\{i = 1\}, y', A_i^T x'\},$$

for given $x' \in \mathcal{M}_i^x$ and $y' \in \mathcal{M}_i^y$. It is easy to see that the server can "open" (make non-zero) the first coordinate in the output for $y$ using $e_1$ and broadcast this progress to other devices. But the question is, how can we unblock the first coordinate for $x$ other coordinates. Now only updates of the type $A_i y'$ or $A_i^T x'$ can help. But since $A_i$ has only $(i-1)$th, $(n+i-1)$th, ... rows of the matrix $A$, to "open" the first coordinate in the output for $x$ we can use $A_2$ matrix (and only it). One can notice that by using $A_2$ we can also "open" the second coordinate for $y$ after "opening" of the first coordinate for $x$. And that's all, to make more progress we need to use $A_3$ and etc.

We come to the conclusion that we constantly have to transfer progress from the machine currently needed (to open the next coordinates for $x$ and then for $y$) to the server and automatically to other machines. According to Definition C.7, a communication round involves either communicating with all devices or only with some uniform and independent batch of devices. The communication round with batch size $b$ of devices is successful (we have the only one needed at the current moment) with probability $\frac{b}{n-1}$. Let $s_1, s_2 \ldots$ times we make communication round with batchsize $1, 2 \ldots$. Then $K = \sum_{j=1}^{n-1} \frac{j}{n-1} s_j$ since in one full package all $n-1$ devices communicate. The total number of successful communication rounds has the sum of binomial distribution with pairs of parameters $(s_j; \frac{j}{n-1})$. Therefore, the expected value of the total number of successful communication rounds is $\sum_{j=1}^{n-1} \frac{j}{n-1} s_j = K$. And then the expectation of the total number of non-zero coordinates in the global output for $y$ is no more than $K$ and for $x$ is $K - 1$ (but we can say that also $K$). This finishes the proof.

$\square$

The remaining idea is standard and has already been encountered in the literature. If we have $k$ non-zero coordinates in the output, then in the best case in these coordinates we guess the solution exactly, in the other coordinates we return zero and make a mistake. Our goal is to estimate the size of this mistake. To do this, we need to understand how the solution of (14) looks like and get the final theorem – see Lemmas C.6, C.7 and Theorem C.8 of [52] for details.

**Theorem C.10** (Theorem 5.6). *Let $\delta, \mu > 0$, $n \in \mathbb{N}$ and $K \in \mathbb{N}$. There exists a distributed saddle point problem. For which the following statements are true:*

- *$f = \frac{1}{n} \sum_{i=1}^{n} f_i : \mathbb{R}^d \times \mathbb{R}^d \to \mathbb{R}$ is $\mu$ - strongly convex–strongly concave,*

- *$\{f_i\}$ are $\delta$-related in mean,*

- *size $d \geq \max\left\{2\log_q\left(\frac{\alpha}{4\sqrt{2}}\right), 2K\right\}$, where $\alpha = \frac{16n\mu^2}{\delta^2}$ and $q = \frac{1}{2}\left(2 + \alpha - \sqrt{\alpha^2 + 4\alpha}\right) \in (0; 1)$,*

- *the solution of the problem is non-zero: $x^* \neq 0$, $y^* \neq 0$.*

*Then for any output $\hat{x}, \hat{y}$ of any method (Definition C.7) with sending $K$ full packages from the devices to the server, one can obtain the following estimate:*

$$\|\hat{x} - x^*\|^2 + \|\hat{y} - y^*\|^2 = \Omega\left(\exp\left(-\frac{16}{1 + \sqrt{\frac{\delta^2}{16\mu^2 n} + 1}} \cdot K\right)\|y_0 - y^*\|^2\right).$$

### C.4   Lower bounds for algorithms with compression

In the proof of these lower bounds, we use the same "bad" problem as in the previous section, but a bit modified definition of methods.

**Definition C.11** (Class of algorithms)**.** Each device $i$ has its own local memories $\mathcal{M}_i^x$ and $\mathcal{M}_i^y$ for the $x$- and $y$-variables –with initialization $\mathcal{M}_m^x = \mathcal{M}_m^y = \{0\}$. $\mathcal{M}_1^x$ and $\mathcal{M}_1^y$ are memories of the server. These memories $\mathcal{M}_m^x$ and $\mathcal{M}_m^x$ can be updated as follows:

- **Communication from devices to server:** During one communication round each device $i$ samples uniformly batch $S_i$ of any size $b$ from $[d]$ and send to the server coordinates of some vector with numbers from $S$, i.e. we add points $x$, $y$ to the local memories of the server according to the next rule:

$$x \in \text{span}\left\{x_1, \bigcup_{i\in[n]}\sum_{l\in S_i}\langle[x_i]_l, e_l\rangle e_l\right\}, \quad y \in \text{span}\left\{y_1, \bigcup_{i\in[n]}\sum_{l\in S_i}\langle[y_i]_l, e_l\rangle e_l\right\},$$

where $x_j \in \mathcal{M}_j^x$ and $y_j \in \mathcal{M}_j^y$. If the batch size is equal to $b$ we say that transmits $b/d$ full packages from the device to the server. Batch of the size $d$ is equal to the situation, when the device sends its memories to the server.

- **Communication from server to devices:** Since in the results for upper bounds we fight for the communications from the devices to the server, we assume that broadcasting from the server to the devices is free and for all $i$ we can add any number of points $x, y$ to the local memories:

$$x \in \text{span}\{x_1, x_i\}, \quad y \in \text{span}\{y_1, y_i\},$$

where $x_j \in \mathcal{M}_j^x$ and $y_j \in \mathcal{M}_j^y$.

- **Local computations:** During local computations each device $i$ can make any computations using $f_i$, i.e. can add points $x, y$ to the local memories:

$$x \in \text{span}\{x', \nabla_x f_i(x'', y'')\}, \quad y \in \text{span}\{y', \nabla_y f_i(x'', y'')\},$$

for given $x', x'' \in \mathcal{M}_i^x$ and $y', y'' \in \mathcal{M}_i^y$.

The final global output is calculated as $\hat{x} \in \mathcal{M}_1^x$, $\hat{y} \in \mathcal{M}_1^y$.

Since here we consider compression regime, we assume that during one round all devices communicate with the server, but with sparsify vectors.

As mentioned above, we consider the same "bad" problem. Hence, Lemma C.8 is valid. We need to check Lemma C.9.

**Lemma C.12.** *Let the problem* (14) + (15) *be solved by any method that satisfies Definition C.7. Then after sending $K$ full packages from the devices to the server, in expectation only the first $K$ coordinates of the final output can be non-zero while the rest of the $d - K$ coordinates are strictly equal to zero.*

**Proof:** There is only one change in the proof. All devices communicate every round. Now we need to randomly select the coordinate we want and then we transmit the progress to the server.

$\square$

Finally, we get the next theorem.

**Theorem C.13.** *Let $\delta, \mu > 0$, $n \in \mathbb{N}$ and $K \in \mathbb{N}$. There exists a distributed saddle point problem. For which the following statements are true:*

- $f = \frac{1}{n}\sum_{i=1}^{n} f_i : \mathbb{R}^d \times \mathbb{R}^d \to \mathbb{R}$ *is $\mu$ - strongly convex–strongly concave,*

- *$\{f_i\}$ are $\delta$-related in mean,*

- *size $d \geq \max\left\{2\log_q\left(\frac{\alpha}{4\sqrt{2}}\right), 2K\right\}$, where $\alpha = \frac{16n\mu^2}{\delta^2}$ and $q = \frac{1}{2}\left(2 + \alpha - \sqrt{\alpha^2 + 4\alpha}\right) \in (0;1)$,*

- *the solution of the problem is non-zero: $x^* \neq 0$, $y^* \neq 0$.*

*Then for any output $\hat{x}, \hat{y}$ of any method (Definition C.11) with sending $K$ full packages from the devices to the server, one can obtain the following estimate:*

$$\|\hat{x} - x^*\|^2 + \|\hat{y} - y^*\|^2 = \Omega\left(\exp\left(-\frac{16}{n + \sqrt{\frac{\delta^2 n}{4\mu^2} + n^2}} \cdot K\right)\|y_0 - y^*\|^2\right).$$

## C.5  Proof of Theorem 5.9

**Lemma C.14.** *Let $\{z^k\}_{k\geq 0}$ be the sequence generated by Algorithm 3 for solving problem (1), which satisfies Assumptions 4.2, 5.7, 5.8. Then we have the following estimate on one iteration:*

$$\mathbb{E}\left[\left\|z^{k+1} - z^*\right\|^2\right] + \mathbb{E}\left[\|m^{k+1} - z^*\|^2\right]$$

$$\leq \left(1 - \tau + p - \frac{\gamma\mu}{2}\right)\mathbb{E}\left[\left\|z^k - z^*\right\|^2\right] + \left(1 + \tau - p - \frac{\gamma\mu}{2}\right)\mathbb{E}\left[\|m^k - z^*\|^2\right]$$

$$+ \left(2 + \frac{4\gamma\tilde{\delta}^2}{\mu} + \frac{4}{\gamma\mu} + 4\gamma^2\tilde{\delta}^2\right)\mathbb{E}\left[\|u^k - \hat{u}^k\|^2\right]$$

$$- \left(1 - \tau - \frac{3\gamma\mu}{2}\right)\mathbb{E}\left[\|z^k - \hat{u}^k\|^2\right] - \left(\tau - \frac{3\gamma\mu}{2} - 4\gamma^2\tilde{\delta}^2\right)\mathbb{E}\left[\|m^k - \hat{u}^k\|^2\right].$$

**Proof:** He we also just need to change a bit the proof of Lemma C.1. In more details, it is enough to replace $\left[\frac{1}{n}\sum_{i=1}^{n} Q_i(F(m^k) - F_1(m^k) - F_i(u_H^k) + F_1(u_H^k))\right]$ in Lemma C.1 with $\left[F_{i^k}(m^k) - F_1(m^k) - F_{i^k}(u^k) + F_1(u^k)\right]$. In Lemma C.1, we need to estimate $\mathbb{E}_Q\left[\frac{1}{n}\sum_{i=1}^{n} Q_i(F_i(m^k) - F_1(m^k) - F_i(u_H^k) + F_1(u_H^k))\right]$, $\mathbb{E}_{Q_i}\left[Q_i(F_i(m^k) - F_1(m^k) - F_i(u_H^k) + F_1(u_H^k))\right]$ and $\left\|\frac{1}{n}\sum_{i=1}^{n} Q_i(F(m^k) - F_1(m^k) - F_i(u_H^k) + F_1(u_H^k))\right\|$, now we consider the following expression:

$$\mathbb{E}_{j_k}\left[\mathbb{E}_Q\left[\frac{1}{n}\sum_{i=1}^{n} Q_i(F_{i,j_k^i}(m^k) - F_{1,j_k^1}(m^k) - F_{i,j_k^i}(u_H^k) + F_{1,j_k^1}(u_H^k))\right]\right]$$

$$= \mathbb{E}_{j_k}\left[\frac{1}{n}\sum_{i=1}^{n}(F_{i,j_k^i}(m^k) - F_{1,j_k^1}(m^k) - F_{i,j_k^i}(u_H^k) + F_{1,j_k^1}(u_H^k))\right]$$

$$= \frac{1}{n}\sum_{i=1}^{n}(F_i(m^k) - F_1(m^k) - F_i(u_H^k) + F_1(u_H^k))$$

$$= F(m^k) - F_1(m^k) - F(u_H^k) + F_1(u_H^k).$$

Here we use the unbiasedness of permutation compressors and of choices of $j_k$. We also need to work with

$$\mathbb{E}\left[\left\|\frac{1}{n}\sum_{i=1}^{n} Q_i(F_{i,j_k^i}(m^k) - F_{1,j_k^1}(m^k) - F_{i,j_k^i}(u_H^k) + F_{1,j_k^1}(u_H^k))\right\|^2\right]$$

$$= \mathbb{E}\left[\left\|\frac{1}{n}\sum_{i=1}^{n} Q_i(F_{i,j_k^i}(m^k) - F_{1,j_k^1}(m^k) - F_{i,j_k^i}(u_H^k) + F_{1,j_k^1}(u_H^k)) - (F(m^k) - F_1(m^k) - F(u_H^k) + F_1(u_H^k))\right\|^2\right]$$

$$+ 2\mathbb{E}\left[\left\langle\frac{1}{n}\sum_{i=1}^{n} Q_i(F_{i,j_k^i}(m^k) - F_{1,j_k^1}(m^k) - F_{i,j_k^i}(u_H^k) + F_{1,j_k^1}(u_H^k)), F(m^k) - F_1(m^k) - F(u_H^k) + F_1(u_H^k)\right\rangle\right]$$

$$- \mathbb{E}\left[\|F(m^k) - F_1(m^k) - F(u_H^k) + F_1(u_H^k)\|^2\right]$$

$$= \mathbb{E}\left[\left\|\frac{1}{n}\sum_{i=1}^{n}Q_i(F_{i,j_k^i}(m^k) - F_{1,j_k^1}(m^k) - F_{i,j_k^i}(u_H^k) + F_{1,j_k^1}(u_H^k)) - (F(m^k) - F_1(m^k) - F(u_H^k) + F_1(u_H^k))\right\|^2\right]$$

$$+ 2\mathbb{E}\left[\left\langle\frac{1}{n}\sum_{i=1}^{n}\mathbb{E}_{j_k}\left[\mathbb{E}_{Q_i}\left[Q_i(F_{i,j_k^i}(m^k) - F_{1,j_k^1}(m^k) - F_{i,j_k^i}(u_H^k) + F_{1,j_k^1}(u_H^k))\right]\right], F(m^k) - F_1(m^k) - F(u_H^k) + F_1(u_H^k)\right\rangle\right]$$

$$- \mathbb{E}\left[\left\|F(m^k) - F_1(m^k) - F(u_H^k) + F_1(u_H^k)\right\|^2\right]$$

$$= \mathbb{E}\left[\left\|\frac{1}{n}\sum_{i=1}^{n}Q_i(F_{i,j_k^i}(m^k) - F_{1,j_k^1}(m^k) - F_{i,j_k^i}(u_H^k) + F_{1,j_k^1}(u_H^k)) - (F(m^k) - F_1(m^k) - F(u_H^k) + F_1(u_H^k))\right\|^2\right]$$

$$+ 2\mathbb{E}\left[\left\langle F(m^k) - F_1(m^k) - F(u_H^k) + F_1(u_H^k), F(m^k) - F_1(m^k) - F(u_H^k) + F_1(u_H^k)\right\rangle\right]$$

$$- \mathbb{E}\left[\left\|F(m^k) - F_1(m^k) - F(u_H^k) + F_1(u_H^k)\right\|^2\right]$$

$$= \mathbb{E}\left[\mathbb{E}_{Q_i}\left[\left\|\frac{1}{n}\sum_{i=1}^{n}Q_i(F_{i,j_k^i}(m^k) - F_{1,j_k^1}(m^k) - F_{i,j_k^i}(u_H^k) + F_{1,j_k^1}(u_H^k)) - (F(m^k) - F_1(m^k) - F(u_H^k) + F_1(u_H^k))\right\|^2\right]\right]$$

$$+ \mathbb{E}\left[\left\|F(m^k) - F_1(m^k) - F(u_H^k) + F_1(u_H^k)\right\|^2\right]$$

$$\leq \frac{1}{n}\sum_{i=1}^{n}\mathbb{E}\left[\left\|F_{i,j_k^i}(m^k) - F_{1,j_k^1}(m^k) - F_{i,j_k^i}(u_H^k) + F_{1,j_k^1}(u_H^k) - (F(m^k) - F_1(m^k) - F_m(u_H^k) + F_1(u_H^k))\right\|^2\right]$$

$$+ \mathbb{E}\left[\left\|F(m^k) - F_1(m^k) - F(u_H^k) + F_1(u_H^k)\right\|^2\right]$$

$$= \frac{1}{n}\sum_{i=1}^{n}\mathbb{E}\left[\left\|F_{i,j_k^i}(m^k) - F_{1,j_k^1}(m^k) - F_{i,j_k^i}(u_H^k) + F_{1,j_k^1}(u_H^k)\right\|^2\right]$$

$$- \frac{2}{n}\sum_{i=1}^{n}\mathbb{E}\left[\left\langle F_{i,j_k^i}(m^k) - F_{1,j_k^1}(m^k) - F_{i,j_k^i}(u_H^k) + F_{1,j_k^1}(u_H^k), F(m^k) - F_1(m^k) - F_m(u_H^k) + F_1(u_H^k)\right\rangle\right]$$

$$+ \frac{1}{n}\sum_{i=1}^{n}\mathbb{E}\left[\left\|F(m^k) - F_1(m^k) - F(u_H^k) + F_1(u_H^k)\right\|^2\right]$$

$$+ \mathbb{E}\left[\left\|F(m^k) - F_1(m^k) - F(u_H^k) + F_1(u_H^k)\right\|^2\right]$$

$$= \frac{1}{n}\sum_{i=1}^{n}\mathbb{E}\left[\mathbb{E}_{j_k}\left[\left\|F_{i,j_k^i}(m^k) - F_{1,j_k^1}(m^k) - F_{i,j_k^i}(u_H^k) + F_{1,j_k^1}(u_H^k)\right\|^2\right]\right]$$

$$- \frac{2}{n}\sum_{i=1}^{n}\mathbb{E}\left[\left\langle\mathbb{E}_{j_k}\left[F_{i,j_k^i}(m^k) - F_{1,j_k^1}(m^k) - F_{i,j_k^i}(u_H^k) + F_{1,j_k^1}(u_H^k)\right], F(m^k) - F_1(m^k) - F_m(u_H^k) + F_1(u_H^k)\right\rangle\right]$$

$$+ 2\mathbb{E}\left[\left\|F(m^k) - F_1(m^k) - F(u_H^k) + F_1(u_H^k)\right\|^2\right]$$

$$= \frac{1}{nM^2}\sum_{i=1}^{n}\sum_{j_1=1}^{M}\sum_{j_2=1}^{M}\mathbb{E}\left[\left\|F_{i,j_1}(m^k) - F_{1,j_2}(m^k) - F_{i,j_1}(u_H^k) + F_{1,j_2}(u_H^k)\right\|^2\right]$$

$$- \frac{2}{n}\sum_{i=1}^{n}\mathbb{E}\left[F_i(m^k) - F_1(m^k) - F_i(u_H^k) + F_1(u_H^k), F(m^k) - F_1(m^k) - F_m(u_H^k) + F_1(u_H^k)\rangle\right]$$

$$+ 2\mathbb{E}\left[\left\|F(m^k) - F_1(m^k) - F(u_H^k) + F_1(u_H^k)\right\|^2\right]$$

$$= \frac{1}{nM^2}\sum_{i=1}^{n}\sum_{j_1=1}^{M}\sum_{j_2=1}^{M}\mathbb{E}\left[\left\|F_{i,j_1}(m^k) - F_{1,j_2}(m^k) - F_{i,j_1}(u_H^k) + F_{1,j_2}(u_H^k)\right\|^2\right].$$

Here we use that all $j_k$ are generated uniformly and independently and also definition of $Q_i$ and Lemma B.1. It remains to use Assumption 5.8 and get

$$\mathbb{E}\left[\left\|\frac{1}{n}\sum_{i=1}^{n}Q_i(F_{i,j_k^i}(m^k) - F_{1,j_k^1}(m^k) - F_{i,j_k^i}(u_H^k) + F_{1,j_k^1}(u_H^k))\right\|^2\right] \leq \tilde{\delta}^2\|m^k - u_H^k\|^2$$

This estimate and (8) have only one difference: $\delta \to \tilde{\delta}$, then (10) is valid for Algorithm 3 with $\tilde{\delta}$. Therefore, the rest of the proof is also correct.

$\square$

**Theorem C.15** (Theorem 5.9). *Let $\{z^k\}_{k\geq 0}$ denote the iterates of Algorithm 3 for solving problem (1), which satisfies Assumptions 4.2, 5.7, 5.8. Then, if we choose the stepsizes $\gamma$, $\eta$ and the momentum $\tau$ as follows:*

$$\tau = p \leq \frac{1}{4}, \quad \gamma = \min\left\{\frac{p}{3\mu}, \frac{\sqrt{p}}{4\tilde{\delta}}, \frac{1}{\tilde{L}}\cdot\left(\frac{H}{12\sqrt{M}\log\frac{40\tilde{L}}{\mu p}} - 1\right)\right\}, \quad \eta = \frac{1}{2\sqrt{M}\left(\tilde{L} + \frac{1}{\gamma}\right)},$$

$$q = \frac{1}{M}, \quad \alpha = 1 - \frac{1}{M}, \quad H \geq 16M\log\frac{40\tilde{L}}{\mu p},$$

*we have the convergence:*

$$\mathbb{E}\left[\|z^K - z^*\|^2\right] \leq \left(1 - \frac{\gamma\mu}{2}\right)^K \cdot 2\|z^0 - z^*\|^2.$$

**Proof:** The loop in line 3 makes $H$ steps of Algorithm 1 from [1]. One can note that the operator in (6) is $\left(\tilde{L} + \frac{1}{\gamma}\right)$-Lipschitz continuous in mean (Assumption 5.7) and $\frac{1}{\gamma}$-strongly monotone (Assumption 4.2). For this setup, we use Theorem 4.9. and Corollary 4.10 of [1] to obtain that

$$\text{with} \quad \eta = \frac{1}{2\sqrt{M}\left(\tilde{L} + \frac{1}{\gamma}\right)}, \quad q = \frac{1}{M}, \quad \alpha = 1 - \frac{1}{M}$$

it holds

$$\|u_H^k - \hat{u}^k\|^2 \leq \max\left\{\exp\left(-\frac{H}{16M}\right); \exp\left(-\frac{\frac{1}{\gamma}\cdot H}{12\sqrt{M}\left(\tilde{L} + \frac{1}{\gamma}\right)}\right)\right\}\cdot 4\|z^k - \hat{u}^k\|^2$$

$$= \max\left\{\exp\left(-\frac{H}{16M}\right); \exp\left(-\frac{H}{12\sqrt{M}(\gamma\tilde{L} + 1)}\right)\right\}\cdot 4\|z^k - \hat{u}^k\|^2.$$

If $\gamma \leq \frac{1}{\tilde{L}}\cdot\left(\frac{H}{12\sqrt{M}\log\frac{40\tilde{L}}{\mu p}} - 1\right)$ and $H \geq 16M\log\frac{40\tilde{L}}{\mu p}$, then one can obtain

$$\|u_H^k - \hat{u}^k\|^2 \leq \frac{\mu p}{40\tilde{L}}\|z^k - \hat{u}^k\|^2.$$

This is exactly the same as what we obtained at the beginning of the proof of Theorem 5.2 (Theorem C.2 in Appendix), and hence since Lemmas C.1 and C.14 differ only by $\delta \to \tilde{\delta}$, we can repeat all the other steps of the proof of Theorem 5.2.

$\square$

### C.6 Proof of Theorem 5.12

The loop in line 3 runs $H$ steps of the Extra Gradient method for the problem:

$$\text{Find } \hat{u}^k \in \mathbb{R}^d \text{ such that } G(\hat{u}^k) = 0,$$

$$\text{with } G(z) = F_1(z) - F_1(m^k) + F(m^k) + \frac{1}{\gamma}(z - z^k - \tau(m^k - z^k)) \tag{16}$$

We start the proof with the following lemma.

**Lemma C.16.** *Let $\{z^k\}_{k\geq 0}$ be the sequence generated by Algorithm 1 for solving problem (1) with $\mathcal{Z} = \mathbb{R}^d$, which satisfies Assumptions 4.1,5.11, 4.3. Then we have the following estimate on one*

**Proof:** We start from line 9. Using that $\mathcal{Z}=\mathbb{R}^d$, we get

$$\mathbb{E}\left[\left\|z^{k+1}-z^*\right\|^2\right]$$
$$=\mathbb{E}\left[\left\|z^k-z^*\right\|^2\right]+2\mathbb{E}\left[\langle z^{k+1}-z^k,z^k-z^*\rangle\right]+\mathbb{E}\left[\left\|z^{k+1}-z^k\right\|^2\right]$$
$$=\mathbb{E}\left[\left\|z^k-z^*\right\|^2\right]+2\mathbb{E}\left[\langle z^{k+1}-z^k,u_H^k-z^*\rangle\right]+2\mathbb{E}\left[\langle z^{k+1}-z^k,z^k-u_H^k\rangle\right]+\mathbb{E}\left[\left\|z^{k+1}-z^k\right\|^2\right]$$
$$=\mathbb{E}\left[\left\|z^k-z^*\right\|^2\right]+2\mathbb{E}\left[\langle z^{k+1}-z^k,u_H^k-z^*\rangle\right]+\mathbb{E}\left[\left\|z^{k+1}-u_H^k\right\|^2\right]-\mathbb{E}\left[\left\|z^k-u_H^k\right\|^2\right]$$
$$=\mathbb{E}\left[\left\|z^k-z^*\right\|^2\right]$$
$$+2\mathbb{E}\left[\langle u_H^k+\gamma\cdot\frac{1}{n}\sum_{i=1}^n Q_i(F_i(m^k)-F_1(m^k)-F_i(u_H^k)+F_1(u_H^k))-z^k,u_H^k-z^*\rangle\right]$$
$$+\mathbb{E}\left[\left\|z^{k+1}-u_H^k\right\|^2\right]-\mathbb{E}\left[\left\|z^k-u_H^k\right\|^2\right]$$
$$=\mathbb{E}\left[\left\|z^k-z^*\right\|^2\right]$$
$$+2\mathbb{E}\left[\langle u_H^k+\gamma\cdot\mathbb{E}_Q\left[\frac{1}{n}\sum_{i=1}^n Q_i(F_i(m^k)-F_1(m^k)-F_i(u_H^k)+F_1(u_H^k))\right]-z^k,u_H^k-z^*\rangle\right]$$
$$+\mathbb{E}\left[\left\|z^{k+1}-u_H^k\right\|^2\right]-\mathbb{E}\left[\left\|z^k-u_H^k\right\|^2\right]$$
$$=\mathbb{E}\left[\left\|z^k-z^*\right\|^2\right]+2\mathbb{E}\left[\langle u_H^k+\gamma\cdot(F(m^k)-F_1(m^k))-z^k,u_H^k-z^*\rangle\right]$$
$$-2\gamma\mathbb{E}\left[\langle F(u_H^k)-F_1(u_H^k),u_H^k-z^*\rangle\right]+\mathbb{E}\left[\left\|z^{k+1}-u_H^k\right\|^2\right]-\mathbb{E}\left[\left\|z^k-u_H^k\right\|^2\right].$$

In the last step, we also use the unbiasedness (Lemma B.1) of permutation compressors and their independence from previous points and iterations (in particular, from $z^k,u_H^k$). With one more auxiliary notation $v^k=z^k+\tau(m^k-z^k)-\gamma\cdot\left(F(m^k)-F_1(m^k)\right)$, we have

$$\mathbb{E}\left[\left\|z^{k+1}-z^*\right\|^2\right] \leq \mathbb{E}\left[\left\|z^k-z^*\right\|^2\right]+2\mathbb{E}\left[\langle u_H^k-v^k,u_H^k-z^*\rangle\right]+\tau\mathbb{E}\left[\langle m^k-z^k,u_H^k-z^*\rangle\right]$$
$$-2\gamma\mathbb{E}\left[\langle F(u_H^k)-F_1(u_H^k),u_H^k-z^*\rangle\right]+\mathbb{E}\left[\left\|z^{k+1}-u_H^k\right\|^2\right]-\mathbb{E}\left[\left\|z^k-u_H^k\right\|^2\right].$$

With Assumption 5.11, we have $\langle F(u_H^k),u_H^k-z\rangle\geq 0$, yields:

$$\mathbb{E}\left[\left\|z^{k+1}-z^*\right\|^2\right] \leq \mathbb{E}\left[\left\|z^k-z^*\right\|^2\right]+\tau\mathbb{E}\left[\langle m^k-z^k,u_H^k-z^*\rangle\right]$$
$$+2\gamma\mathbb{E}\left[\langle F_1(u_H^k)+\frac{1}{\gamma}(u_H^k-v^k),u_H^k-z^*\rangle\right]$$
$$+\mathbb{E}\left[\left\|z^{k+1}-u_H^k\right\|^2\right]-\mathbb{E}\left[\left\|z^k-u_H^k\right\|^2\right].$$

The equality $2\langle a,b\rangle=\|a+b\|^2-\|a\|^2-\|b\|^2$ gives

$$\mathbb{E}\left[\left\|z^{k+1}-z^*\right\|^2\right] \leq \mathbb{E}\left[\left\|z^k-z^*\right\|^2\right]+2\gamma\mathbb{E}\left[\langle F_1(u_H^k)+\frac{1}{\gamma}(u_H^k-v^k),u_H^k-z^*\rangle\right]$$
$$+\tau\mathbb{E}\left[\langle m^k-u_H^k,u_H^k-z^*\rangle\right]+\tau\mathbb{E}\left[\langle u_H^k-z^k,u_H^k-z^*\rangle\right]$$
$$+\mathbb{E}\left[\left\|z^{k+1}-u_H^k\right\|^2\right]-\mathbb{E}\left[\left\|z^k-u_H^k\right\|^2\right]$$

$$
\begin{aligned}
=&\mathbb{E}\left[\left\|z^k - z^*\right\|^2\right] + 2\gamma\mathbb{E}\left[\left\langle F_1(u_H^k) + \frac{1}{\gamma}(u_H^k - v^k), u_H^k - z^*\right\rangle\right] \\
&+ \tau\mathbb{E}\left[\|m^k - z^*\|^2\right] - \tau\mathbb{E}\left[\|m^k - u_H^k\|^2\right] - \tau\|u_H^k - z^*\|^2 \\
&+ \tau\mathbb{E}\left[\|u_H^k - z^k\|^2\right] + \tau\mathbb{E}\left[\|u_H^k - z^*\|^2\right] - \tau\mathbb{E}\left[\|z^k - z^*\|^2\right] \\
&+ \mathbb{E}\left[\left\|z^{k+1} - u_H^k\right\|^2\right] - \mathbb{E}\left[\left\|z^k - u_H^k\right\|^2\right] \\
=&(1-\tau)\mathbb{E}\left[\left\|z^k - z^*\right\|^2\right] + \tau\mathbb{E}\left[\|m^k - z^*\|^2\right] \\
&+ 2\gamma\mathbb{E}\left[\left\langle F_1(u_H^k) + \frac{1}{\gamma}(u_H^k - v^k), u_H^k - z^*\right\rangle\right] \\
&+ \mathbb{E}\left[\left\|z^{k+1} - u_H^k\right\|^2\right] - (1-\tau)\mathbb{E}\left[\left\|z^k - u_H^k\right\|^2\right] - \tau\mathbb{E}\left[\|m^k - u_H^k\|^2\right].
\end{aligned}
$$

By Cauchy Schwartz inequality $2\langle a, b\rangle \le \|a\| \cdot \|b\|$, we have

$$
\begin{aligned}
\mathbb{E}\left[\left\|z^{k+1} - z^*\right\|^2\right] \le&(1-\tau)\mathbb{E}\left[\left\|z^k - z^*\right\|^2\right] + \tau\mathbb{E}\left[\|m^k - z^*\|^2\right] \\
&+ 2\gamma\mathbb{E}\left[\left\|F_1(u_H^k) + \frac{1}{\gamma}(u_H^k - v^k)\right\| \cdot \|u_H^k - z^*\|\right] \\
&+ \mathbb{E}\left[\left\|z^{k+1} - u_H^k\right\|^2\right] - (1-\tau)\mathbb{E}\left[\left\|z^k - u_H^k\right\|^2\right] - \tau\mathbb{E}\left[\|m^k - u_H^k\|^2\right] \\
=&(1-\tau)\mathbb{E}\left[\left\|z^k - z^*\right\|^2\right] + \tau\mathbb{E}\left[\|m^k - z^*\|^2\right] \\
&+ 2\gamma\mathbb{E}\left[\left\|F_1(u_H^k) + \frac{1}{\gamma}(u_H^k - v^k)\right\| \cdot \|u_H^k - z^*\|\right] \\
&+ \gamma^2\mathbb{E}\left[\left\|\frac{1}{n}\sum_{i=1}^n Q_i(F_i(m^k) - F_1(m^k) - F_i(u_H^k) + F_1(u_H^k))\right\|^2\right] \\
&- (1-\tau)\mathbb{E}\left[\left\|z^k - u_H^k\right\|^2\right] - \tau\mathbb{E}\left[\|m^k - u_H^k\|^2\right]. \quad\quad (17)
\end{aligned}
$$

One can substitute (8) in (17) and get

$$
\begin{aligned}
\mathbb{E}\left[\left\|z^{k+1} - z^*\right\|^2\right] \le&(1-\tau)\mathbb{E}\left[\left\|z^k - z^*\right\|^2\right] + \tau\mathbb{E}\left[\|m^k - z^*\|^2\right] \\
&+ 2\gamma\mathbb{E}\left[\left\|F_1(u_H^k) + \frac{1}{\gamma}(u_H^k - v^k)\right\| \cdot \|u_H^k - z^*\|\right] \\
&+ 2\gamma^2\delta^2\mathbb{E}\left[\left\|m^k - u_H^k\right\|^2\right] - (1-\tau)\mathbb{E}\left[\left\|z^k - u_H^k\right\|^2\right] - \tau\mathbb{E}\left[\|m^k - u_H^k\|^2\right] \\
\le&(1-\tau)\mathbb{E}\left[\left\|z^k - z^*\right\|^2\right] + \tau\mathbb{E}\left[\|m^k - z^*\|^2\right] \\
&+ 2\gamma\mathbb{E}\left[\left\|F_1(u_H^k) + \frac{1}{\gamma}(u_H^k - v^k)\right\| \cdot \|u_H^k - z^*\|\right] \\
&- (\tau - 2\gamma^2\delta^2)\mathbb{E}\left[\left\|m^k - u_H^k\right\|^2\right] - (1-\tau)\mathbb{E}\left[\left\|z^k - u_H^k\right\|^2\right]. \quad\quad (18)
\end{aligned}
$$

Then, summing up (18) and (11), we have

$$
\begin{aligned}
\mathbb{E}\left[\left\|z^{k+1} - z^*\right\|^2 + \|m^{k+1} - z^*\|^2\right] \le&(1-\tau+p)\mathbb{E}\left[\left\|z^k - z^*\right\|^2\right] + (1+\tau-p)\mathbb{E}\left[\|m^k - z^*\|^2\right] \\
&+ 2\gamma\mathbb{E}\left[\left\|F_1(u_H^k) + \frac{1}{\gamma}(u_H^k - v^k)\right\| \cdot \|u_H^k - z^*\|\right] \\
&- (\tau - 2\gamma^2\delta^2)\mathbb{E}\left[\|m^k - u_H^k\|^2\right] - (1-\tau)\mathbb{E}\left[\left\|z^k - u_H^k\right\|^2\right].
\end{aligned}
$$

$\square$

Now we are ready to prove the theorem.

**Theorem C.17** (Theorem 5.12)**.** *Let $\{z^k\}_{k\geq 0}$ denote the iterates of Algorithm 1 for solving problem (1) with $\mathcal{Z} = \mathbb{R}^d$, which satisfies Assumptions 4.1, 5.11, 4.3. Then, if we additionally assume that $\|z^k\|, \|u_H^k\|, \|z^*\| \leq \Omega$ and choose the stepsizes $\gamma$, $\eta$ and the momentum $\tau$ as follows:*

$$\tau = p \leq \frac{1}{2}, \quad \gamma = \frac{\sqrt{p}}{6\delta}, \tag{19}$$

*we have the convergence:*

$$\frac{1}{K}\sum_{k=0}^{K-1} \mathbb{E}\left[\|F(u_H^k)\|^2\right] \lesssim \frac{\delta^2 \Omega^2}{pK}\left(\frac{1}{K} + \frac{1}{\sqrt{H}}\left(\frac{L\sqrt{p}}{6\delta} + 1\right) + \frac{1}{H}\left(\frac{L\sqrt{p}}{6\delta} + 1\right)^2\right).$$

**Proof:** The loop in line 3 makes $H$ steps of the Extra Gradient method for the problem (16). Note that the operator in (16) is $\left(L + \frac{1}{\gamma}\right)$-Lipschitz continuous (Assumption 4.1). For this case, there are convergence guarantees in the literature (see Theorem 4.1 from [33]):

$$\text{with} \quad \eta = \frac{1}{2\left(L + \frac{1}{\gamma}\right)} \quad \text{it holds} \quad \left\|F_1(u_H^k) + \frac{1}{\gamma}(u_H^k - v^k)\right\|^2 \leq \frac{28}{\eta^2 H}\left\|z^k - \hat{u}^k\right\|^2$$

$$= \frac{112}{H}\left(L + \frac{1}{\gamma}\right)^2 \left\|z^k - \hat{u}^k\right\|^2. \tag{20}$$

Combining this fact and Lemma C.16 gives

$$\mathbb{E}\left[\left\|z^{k+1} - z^*\right\|^2 + \|m^{k+1} - z^*\|^2\right] \leq \mathbb{E}\left[\left\|z^k - z^*\right\|^2\right] + \mathbb{E}\left[\|m^k - z^*\|^2\right]$$

$$+ \frac{24\gamma}{\sqrt{H}}\left(L + \frac{1}{\gamma}\right)\mathbb{E}\left[\|z^k - \hat{u}^k\| \cdot \|u_H^k - z^*\|\right]$$

$$- (\tau - 2\gamma^2\delta^2)\mathbb{E}\left[\|m^k - u_H^k\|^2\right] - (1 - \tau)\mathbb{E}\left[\left\|z^k - u_H^k\right\|^2\right].$$

With $\|z^k\|, \|u_H^k\|, \|\hat{u}^k\|, \|z^*\| \leq \Omega$, one can obtain

$$\mathbb{E}\left[\left\|z^{k+1} - z^*\right\|^2 + \|m^{k+1} - z^*\|^2\right] \leq \mathbb{E}\left[\left\|z^k - z^*\right\|^2\right] + \mathbb{E}\left[\|m^k - z^*\|^2\right] + \frac{100\Omega^2}{\sqrt{H}}(\gamma L + 1)$$

$$- (1 - \tau)\mathbb{E}\left[\left\|z^k - u_H^k\right\|^2\right] - (\tau - 2\gamma^2\delta^2)\mathbb{E}\left[\|m^k - u_H^k\|^2\right].$$

Making small rearrangements, we get

$$(1 - \tau)\mathbb{E}\left[\left\|z^k - u_H^k\right\|^2\right] \leq \mathbb{E}\left[\left\|z^k - z^*\right\|^2 + \|m^k - z^*\|^2\right] - \mathbb{E}\left[\left\|z^{k+1} - z^*\right\|^2 + \|m^{k+1} - z^*\|^2\right]$$

$$+ \frac{100\Omega^2}{\sqrt{H}}(\gamma L + 1) - (\tau - 2\gamma^2\delta^2)\mathbb{E}\left[\|m^k - u_H^k\|^2\right].$$

Summing over all $k$ from 0 to $K - 1$ and dividing by $K$, we have

$$\frac{(1 - \tau)}{K}\sum_{k=0}^{K-1}\mathbb{E}\left[\left\|z^k - u_H^k\right\|^2\right] \leq \frac{\left[\left\|z^0 - z^*\right\|^2 + \|m^0 - z^*\|^2\right]}{K} + \frac{100\Omega^2}{\sqrt{H}}(\gamma L + 1)$$

$$- (\tau - 2\gamma^2\delta^2)\frac{1}{K}\sum_{k=0}^{K-1}\mathbb{E}\left[\|m^k - u_H^k\|^2\right]. \tag{21}$$

Next, we work with $\frac{1}{K}\sum_{k=0}^{K-1}\mathbb{E}\left[\left\|z^k - u_H^k\right\|^2\right]$. With notation $v^k = z^k + \tau(m^k - z^k) - \gamma \cdot \left(F(m^k) - F_1(m^k)\right)$, we have

$$\frac{1}{K}\sum_{k=1}^{K}\left\|z^k - u_H^k\right\|^2 = \frac{1}{K}\sum_{k=1}^{K}\left\|v^k - \tau(m^k - z^k) + \gamma\left(F(m^k) - F_1(m^k)\right) - u_H^k\right\|^2$$

$$= \frac{1}{K}\sum_{k=1}^{K}\left\|v^k - \tau(m^k - u_H^k + u_H^k - z^k) + \gamma\left(F(m^k) - F_1(m^k)\right)\right.$$

$$\left. - \gamma\left(F(u_H^k) - F_1(u_H^k)\right) + \gamma\left(F(u_H^k) - F_1(u_H^k)\right) - u_H^k\right\|^2.$$

Applying Young's inequality $\|a+b\|^2 \leq 2\|a\|^2 + 2\|b\|^2$ gives

$$\frac{1}{K}\sum_{k=1}^{K}\left\|z^k - u_H^k\right\|^2 \geq \frac{\gamma^2}{2K}\sum_{k=1}^{K}\|F(u_H^k)\|^2 - \frac{1}{K}\sum_{k=1}^{K}\| -\gamma F_1(u_H^k) + v^k - u_H^k - \tau(m^k - u_H^k)$$

$$- \tau(u_H^k - z^k) + \gamma\left(F(m^k) - F_1(m^k)\right) - \gamma\left(F(u_H^k) - F_1(u_H^k)\right)\|^2$$

$$\geq \frac{\gamma^2}{2K}\sum_{k=1}^{K}\|F(u_H^k)\|^2 - \frac{4}{K}\sum_{k=1}^{K}\| -\gamma F_1(u_H^k) + v^k - u_H^k\|^2$$

$$- \frac{4\tau^2}{K}\sum_{k=1}^{K}\|m^k - u_H^k\|^2 - \frac{4\tau^2}{K}\sum_{k=1}^{K}\|u_H^k - z^k\|^2$$

$$- \frac{4\gamma^2}{K}\sum_{k=1}^{K}\|\left(F(m^k) - F_1(m^k)\right) - \left(F(u_H^k) - F_1(u_H^k)\right)\|^2.$$

After small rearrangements we have

$$\frac{1+4\tau^2}{K}\sum_{k=1}^{K}\left\|z^k - u_H^k\right\|^2 \geq \frac{\gamma^2}{2K}\sum_{k=1}^{K}\|F(u_H^k)\|^2 - \frac{4}{K}\sum_{k=1}^{K}\| -\gamma F_1(u_H^k) + v^k - u_H^k\|^2$$

$$- \frac{4\gamma^2}{K}\sum_{k=1}^{K}\|\left(F(m^k) - F_1(m^k)\right) - \left(F(u_H^k) - F_1(u_H^k)\right)\|^2$$

$$- \frac{4\tau^2}{K}\sum_{k=1}^{K}\|m^k - u_H^k\|^2. \tag{22}$$

Then, with combining (21) and (22), we obtain

$$\frac{(1-\tau)\gamma^2}{2K}\sum_{k=0}^{K-1}\mathbb{E}\left[\|F(u_H^k)\|^2\right] \leq \frac{(1+4\tau^2)\left[\|z^0 - z^*\|^2 + \|m^0 - z^*\|^2\right]}{K} + \frac{100\Omega^2}{\sqrt{H}}(\gamma L + 1)(1 + 4\tau^2)$$

$$- (1+4\tau^2)(\tau - 2\gamma^2\delta^2)\frac{1}{K}\sum_{k=0}^{K-1}\mathbb{E}\left[\|m^k - u_H^k\|^2\right]$$

$$+ \frac{4(1-\tau)}{K}\sum_{k=1}^{K}\| -\gamma F_1(u_H^k) + v^k - u_H^k\|^2$$

$$+ \frac{4\tau^2(1-\tau)}{K}\sum_{k=1}^{K}\|m^k - u_H^k\|^2$$

$$+ \frac{4(1-\tau)\gamma^2}{K}\sum_{k=1}^{K}\|\left(F(m^k) - F_1(m^k)\right) - \left(F(u_H^k) - F_1(u_H^k)\right)\|^2.$$

Using (20) and Assumption 4.3, we get

$$\frac{(1-\tau)\gamma^2}{2K}\sum_{k=0}^{K-1}\mathbb{E}\left[\|F(u_H^k)\|^2\right] \leq \frac{(1+4\tau^2)\left[\|z^0 - z^*\|^2 + \|m^0 - z^*\|^2\right]}{K} + \frac{100\Omega^2}{\sqrt{H}}(\gamma L + 1)(1 + 4\tau^2)$$

$$- (1+4\tau^2)(\tau - 2\gamma^2\delta^2)\frac{1}{K}\sum_{k=0}^{K-1}\mathbb{E}\left[\|m^k - u_H^k\|^2\right]$$

$$+ \frac{2000(1-\tau)}{H}(\gamma L + 1)^2\Omega^2$$

$$+ \frac{4\tau^2(1-\tau)}{K}\sum_{k=1}^{K}\|m^k - u_H^k\|^2$$

$$+ \frac{4(1-\tau)\gamma^2\delta^2}{K}\sum_{k=1}^{K}\|m^k - u_H^k\|^2$$

$$= \frac{(1 + 4\tau^2) \left[ \left\| z^0 - z^* \right\|^2 + \left\| m^0 - z^* \right\|^2 \right]}{K} + \frac{100\Omega^2}{\sqrt{H}} (\gamma L + 1)(1 + 4\tau^2)$$

$$- \left[ \tau(1 - \tau + 8\tau^2) - 2\gamma^2(6 - 4\tau + 8\tau^2)\delta^2 \right] \frac{1}{K} \sum_{k=0}^{K-1} \mathbb{E} \left[ \| m^k - u_H^k \|^2 \right]$$

$$+ \frac{2000(1 - \tau)}{H} (\gamma L + 1)^2 \Omega^2.$$

With the choice $\tau$ from (19), we obtain

$$\frac{\gamma^2}{4K} \sum_{k=0}^{K-1} \mathbb{E} \left[ \| F(u_H^k) \|^2 \right] \leq \frac{2 \left[ \left\| z^0 - z^* \right\|^2 + \left\| m^0 - z^* \right\|^2 \right]}{K} + \frac{200\Omega^2}{\sqrt{H}} (\gamma L + 1)$$

$$- \left( \frac{\tau}{2} - 16\gamma^2\delta^2 \right) \frac{1}{K} \sum_{k=0}^{K-1} \mathbb{E} \left[ \| m^k - u_H^k \|^2 \right] + \frac{2000}{H} (\gamma L + 1)^2 \Omega^2.$$

Additionally, with the choice $\gamma$ from (19), we get

$$\frac{1}{K} \sum_{k=0}^{K-1} \mathbb{E} \left[ \| F(u_H^k) \|^2 \right] \leq \frac{300\delta^2 \left[ \left\| z^0 - z^* \right\|^2 + \left\| m^0 - z^* \right\|^2 \right]}{pK}$$

$$+ \frac{3 \cdot 10^4 \cdot \delta^2 \Omega^2}{p\sqrt{H}} \left( \frac{L\sqrt{p}}{6\delta} + 1 \right) + \frac{3 \cdot 10^5 \cdot \delta^2 \Omega^2}{Hp} \left( \frac{L\sqrt{p}}{6\delta} + 1 \right)^2.$$

It remains to take into account that $m^0 = z^0$.

$\square$