# OpenReview forum: "Similarity, Compression and Local Steps: Three Pillars of Efficient Communications for Distributed Variational Inequalities"
_NeurIPS.cc/2023/Conference — NeurIPS 2023 poster_

### Official Review · Reviewer_JXTW · 2023-06-26

**Soundness:** 3 good
**Presentation:** 1 poor
**Contribution:** 2 fair
**Rating:** 6
**Confidence:** 3

**Summary:**

This paper investigates distributed variational inequalities (VI) problems in the context of a server-workers architecture. It proposes a method that combines three major communication-saving techniques: i) compression, ii) local steps, and iii) data similarity across workers. For strongly monotone VI and strongly-convex strongly-concave saddle point problems (SPP), the proposed method outperforms competing methods in terms of communication complexity. Furthermore, it improves on previous results for non-monotone operators. It also extends the technique to a partial participation setting, in which one random device is selected at each iteration to send an uncompressed parameter while all other devices remain idle.

**Strengths:**

This paper employs three communication-saving techniques (local methods, compression, and similarity) to achieve the best theoretical communication complexity for distributed methods solving strongly-monotone VI problems.

**Weaknesses:**

- I believe that the writing and organization could be much improved. The paper is not well explained.  There several typos and unclear statements. (line 172, the inner product in assumption 4.2 a semicolon is used instead of comma) (Missing comma in line 174 after i.e.) (initial value of $m^k$ is not specified in Alg 1) (line 10 in alg 1 $z^{k+1}$ instead of $z^k$) (definition of communication complexity is unclear and uncommon) (line 203 ‘we’ is missing an e) (having the main computation on server is not a contribution since there are methods with the same feature)

The description of the algorithm is too long and not organized well. In page 6, lines 228—229 the author describe a reformulation that I find unclear where it came from. Moreover, in this part, variables $m^k,v^k$ are introduced where $m^k$ is a reference sequence that is only explained after several paragraphs. Similarly for the momentum term.

The contribution is incremental in my opinion and builds on prior works [12] and [16]. The result is also a minor improvement over [51]. Under strong monotonicity the analysis is not that challenging and follows by using basic techniques from the literature. Moreover, assumption 4.3 is a strong assumption since data heterogeneity is common in practice. This is not worth a NeuRIPS paper on its own.

The algorithm does not use full compression since it occasionally sends uncompressed vector in uplink (with small probability). Also it does not use compression in the downlink and it is unclear how to do that (even though the authors mention that their method can be generalized to two-sided compression).


**Questions:**

The lower bound mentioned in Table 1 does not consider compression and, in my opinion, appears irrelevant. Could you please provide a lower bound that takes compression into account? This would be valuable in assessing the optimality of the proposed method. I understand that this aspect is currently designated for future research, but I strongly believe that it would be a significant contribution that enhances the paper

Could it be said that permutation compressors are more accurately characterized as a sparsification technique rather than a compression method?

In the context of this study, communication complexity is defined as the aggregate information transmitted. Specifically, in line 283, it is indicated that an average of $O(p+1/n)$ information is communicated. I would greatly appreciate a precise explanation of the concept of "information" within this context. Furthermore, I would appreciate more clarification regarding the division of communication rounds by $n$. Is this division influenced by compression techniques, such as sparsification, where $p$ corresponds to $1/n$?
I would like to inquire about the validity of setting $H$ to 1 in line 296. In such a scenario, $\gamma$ would be equal to 0, resulting in an undefined bound. Could you please provide clarification regarding this aspect?

Is the probability $p$ required to be: $0 < p < 1$?

I would appreciate clarification regarding Theorem 5.7, as I find it confusing. How is it possible for $H$ to be of the order $O(K^2)$? Multiplying the communication rounds rate by $H$ seems to result in a computational bound that increases with $K$. Could you please provide further explanation on this matter?


**Limitations:**

No. For example, the studied assumptions are strict and not practical.

---

> ### Author Rebuttal · Authors · 2023-08-08
>
> We thank Reviewer __JXTW__ for the time, work and detailed review of our paper! Next, we will answer the issues raised.
>
> But before the response, we note that __we added new results to the revised paper__ - see our general response! Reviewer __JXTW__ requested __lower bounds__. We made them and showed that __our result is optimal and unimprovable!__ Reviewers __D8HM__ and __4qia__ asked to extend the results to __the stochastic case. This we also added!__(see the response to Reviewer __4qia__). We hope that these results will further enhance the paper's contribution!
>
> __W1__(writing):
>
> Many thanks for the attentive work with our paper!
> 1) We independently proofread and revised the paper and tried to eliminate the problems not only put forward by the reviewers
> 2) We added $m^0 =z^0$
> 3) line 10 in alg 1 $z^{k+1}$ instead of $z^k$: this is not a typo, we really need $z^k$ (not $z^{k+1}$), we use $z^{k}$ in the proof (lines 638-639), we found it more convenient for proofs. It doesn't make much difference, we can change it to $z^{k+1}$, but then we have to change the proofs a bit
> 4) communication complexity: The reviewer further clarifies the issue and asks what we understand by "transmitted information". Our response: due to the nature of our compressors (as the reviewer correctly notes we use a sparsifier, which reduces the number of transmitted coordinates), by "transmitted information" we mean the number of coordinates transmitted - we added this to line 195 in the revision
> 5) We added in the paragraph "Main calculations on server" a mention of other works for VIs that also have this property. But we can remove this paragraph at all if it seems irrelevant
>
> __W2__(description):
>
> In line 224 we give the idea the local problem reguralization. There it is given for the minimization problem $\min_x [f_1(x)+\lambda ||x-x_s||^2]$. In Section 4 (lines 160-162) we give that the minimization problem can be written as a variational inequality with $F=\nabla f$. Then in the case of $f(z)=f_1(z)+\lambda ||z-z_s||^2$ we can take $F(z)=\nabla f_1(z)+\lambda (z-z_s)=F_1(z)+\lambda (z-z_s)$. This is what we have in line 229. We added a bit more explanation to the revised paper to avoid misunderstandings. We also added that the nature of the $v_k$ point will be defined later in the text
>
> __W3__(contribution):
>
> 1) We agree that the assumption on data similarity is a bit limiting, but it is quite popular in the literature. In the response to Reviewer __kubg__ we cite papers from top conferences where similarity was investigated [R1-11]. All these papers consider (strongly) convex/monotone problems
> 2) It is a known fact that the non-smooth non-convex problem is NP-hard [R1]. Acceptable theoretical estimates of convergence can be obtained only by introducing assumptions about the problem. At the same time, the same NNs, being non-smooth non-convex, often behave as convex problems in the process of training. Therefore, considering even convex problems can give a boost in terms of understanding optimization methods
> 3) No one before this paper has combined all three techniques even for minimization problems. Yes, similarity for SPPs/VIs has been considered before us. Yes, some results (including with compression) have been done before. But the record and (judging by the lower bounds) unimprovable result is obtained here. This paper was able to unite all the past ideas and achieve optimality
> 4) As we mentioned, [12,16,51] consider only monotone case, we also consider non-monotone case (Sec. 5.4)
>
> [R1] Kornowski et al. Oracle complexity in nonsmooth nonconvex optimization
>
> __W4__(compression):
>
> Yes, these two things are the limitations of our algorithm, the paper doesn't hide it. Note that the setup when only one-side communication is used is really popular and most of the works on compression stop on this setting (see e.g. works from Table 1 from [R1], we also have motivation of it - see lines 202-203). Note also that people write separate papers to a) add two-side compression ([R1] $\to$ [R2]) or b) remove rare sending of full packages ([R3] $\to$ [R4])
>
> [R1] Richtárik et al. EF21: A new, simpler, theoretically better, and practically faster error feedback __NeurIPS__
>
> [R2] Fatkhullin et al. EF21 with bells whistles: Practical algorithmic extensions of modern error feedback
>
> [R3] Gorbunov et al. MARINA: Faster non-convex distributed learning with compression __ICML__
>
> [R4] Tyurin et al. DASHA: Distributed nonconvex optimization with communication compression, optimal oracle complexity __ICLR__
>
> __Q1__(lower bounds):
>
> See our general reply
>
> __Q2__(compression):
>
> Yes, PermK is a sparsifier but the original paper [80] calls it compression. Moreover, in the literature sparsifiers are often considered as a subset of compressors
>
> __Q3__(complexity):
>
> Due to the nature of our compressors, by "transmitted information" we mean the number of coordinates transmitted - we added this to line 195 in the revision.  With probability $p$ we send a full vector with $d$ coordinates from the devices to the server.  And if we use PermK compression, we forward $d/n$ coordinates. Then the average number of coordinates from the devices to the server is equal to $dp+d/n=\mathcal{O}(p+1/n)$. We added this explanation
>
> About $H$: it is a typo, we corrected this and wrote $H=\mathcal{O}(1)$ in lines 296 and 298
>
> __Q4__($p$):
>
> Yes, $p\in (0;1]$. We added this to the text of the revised paper and to Algorithm 1
>
> __Q5__(Th. 5.7):
>
> The computational complexity can be found as $T=H\cdot K=O(K^3)$. Also from Th. 5.7 (as in Cor. 5.8), the iterative complexity $K=O(1/\varepsilon^2)$ can be found. Using this $K$, we can find $T=O(1/\varepsilon^6)$.
> Perhaps the reviewer would like us to leave the results of Th. 5.7 without choosing $H$. Then in Th. 5.7 we would have $||F(u)||^2= O(1/K+1/\sqrt{H})$. Using this expression, we can find that $K\sim O(1/\varepsilon^2)$ and $H\sim O(1/\varepsilon^4)$ and $T=O(1/\varepsilon^6)$

---

> > ### Comment · Reviewer_JXTW · 2023-08-11
> >
> > I've gone through the rebuttal, and the authors have taken into account several of my comments. As a result, I have adjusted my evaluation accordingly.

---

> > > ### Author Response · Authors · 2023-08-12
> > >
> > > Thank you so much! And thanks again for the very detailed review! It has helped and is helping to make our paper better!

---

### Official Review · Reviewer_4qia · 2023-06-26

**Soundness:** 4 excellent
**Presentation:** 4 excellent
**Contribution:** 3 good
**Rating:** 7
**Confidence:** 3

**Summary:**

The paper presents a study of distributed algorithms for solving variational inequalities in a centralized distributed optimization setting. The authors propose three improvements: compression, data similarity, and local steps. Compression reduces the communication complexity by transmitting compressed versions of the data, data similarity reduces the variance among workers by sending batches of data that are similar across workers, and local steps reduce the communication rounds by allowing each worker to perform multiple iterations locally before communicating with the central server. The paper's main contribution is achieving the best known theoretical convergence compared to other deterministic algorithms for solving variational inequalities bot for monotone and strongly monotone VI.

**Strengths:**

The paper's major strength lies in its successful implementation of three known techniques to improve communication complexity in distributed learning, combined in a way that maximizes their effectiveness. The authors pushed these techniques to their limits and studied the best theoretical combination, which is a significant contribution to the field. While the techniques themselves are not novel, the authors' implementation and optimization sets their work apart. Additionally, the paper provides a thorough analysis of the convergence and complexity of the proposed algorithm, which adds to its strength.

**Weaknesses:**

However, a potential weakness of the paper is the centralized distributed optimization setting considered by the authors. The paper assumes the presence of a central server that synchronizes the work among the different workers, which may not always be feasible in real-life applications.

Moreover, the author should address synchronization issues, such as a worker taking too long to send their result to the server. The paper could benefit from more discussion on the limitations and applicability of the proposed algorithm in different distributed settings.

**Questions:**

The paper is clear and well-written, and the proposed algorithm's convergence and complexity are thoroughly analyzed. While I don't have any specific questions, I believe it would be helpful to know how the proposed algorithm performs in other distributed settings and to what extent the synchronization assumptions affect its performance, especially in the case of communication delays or worker failures. Providing insights on these aspects would enhance the applicability of the proposed algorithm and help readers understand its potential limitations.

**Limitations:**

In addition to the synchronization issue, I believe that a stochastic version of the algorithm would be more appropriate for handling large-scale distributed optimization problems, as mentioned by the authors.

Moreover, the limitations are not sufficiently discussed by the authors.

---

> ### Author Rebuttal · Authors · 2023-08-08
>
> We thank Reviewer __4qia__ for the work, time, and positive feedback! Below we will respond to the issues raised.
>
> But before the response, we note that __we added new results to the new version of the paper - see our general response!__  We added __the extension of our algorithms to the stochastic case__ (as requested by Reviewer __4qia__ in _Limitations_) - see the second part of this response. Reviewer __JXTW__ requested __lower bounds__. We made them and showed that __our result is optimal and unimprovable!__ We hope that these results will further enhance the paper's contribution!
>
> __W__ and __Q__(server and synchronization):
>
> Thank you! This is really important!
>
> We think that the answers to these issues have already been partially reflected in the original version of the papers (page 7 lines 307-314). In more detail,
>
> 1) _(Server)_  (See lines 307 - 311) The existence of a central server to which all devices are attached is nominal. In fact, all devices can be equivalent and connected in a network, for example, in the form of a ring. We just need to select a nominal server. It is enough for us that at the right moments, all devices send messages to the "server", and the server then sends them responses. For this purpose, in decentralized networks we can apply various AllReduce/AllGather/etc procedures.
>
> 2) _(Synchronization)_ (See lines 311 - 314) Here we honestly report that our Algorithm 1 does not support asynchronous communications, nor communications delays. And Algorithm 2 is proposed to partially solve this issue because in Algorithm 2, only 1 device communicates each iteration.
>
> __Stochastic case__
>
> Consider the following stochastic setting. Let $f_i$s have the sum type:
> $$
> F_i = \frac{1}{M} \sum\limits_{j=1}^M F_{i, j}.
> $$
> And we assume that we can now call not the full operators $F_i$, but only the operators by batches $F_{i,j}$. Such a setup is more than natural especially in the context of similarity. For example, if we initially have a large dataset $N = nMb$ and distribute it to $n$ devices, then each device has $Mb$ samples:
> $$
> F = \frac{1}{n} \sum\limits_{i=1}^n F_i = \frac{1}{n} \sum\limits_{i=1}^n \left[ \frac{1}{M} \sum\limits_{j=1}^M F_{i,j}\right] = \frac{1}{n} \sum\limits_{i=1}^n \left[ \frac{1}{M} \sum\limits_{j=1}^M  \left(  \frac{1}{b} \sum\limits_{l=1}^b F_{i,j,l}\right) \right]
> $$
> The nature of similarity goes precisely that $F_i$ have the form of a sub-sum (taken from a common large sum). The larger the sample size $Mb$ for $F_i$, the smaller the similarity parameter $\delta$. Now in the stochastic case we consider $F_{i,j}$ and they also have similarity, but with a larger similarity parameter $\tilde \delta$ (since $Mb$ is larger than $b$). In particular, we use
>
> _Assumption 5.9_
> For all $i_1, i_2 \in 1,\ldots n$, $j \in 1,\ldots M$ and $u, v \in Z$
> $$
> \frac{1}{M} \sum\limits_{j_2 =1}^M || F_{i_1,j_1}(u) - F_{i_2, j_2}(u)- F_{i_1,j_1}(v) + F_{i_2, j_2}(v)||^2 \leq \tilde \delta^2 ||u - v||^2
> $$
>
> We modify Algorithm 1 as follows:
>
> 1) Line 7: Server broadcasts $u^k_H$ and $F_{1, j_k^1}(u^k_H) - F_{1, j_k^1}(m^k)$ to devices, where $j_k^1$ is generated uniformly from $[1, \ldots, M]$
>
> 2) Line 8: Devices in parallel compute $Q_i(F_{i, j_k^i}(m^k) - F_{1, j_k^1}(m^k) - F_{i, j_k^i}(u^k_H) + F_{1, j_k^1}(u^k_H))$, where $j_k^i$ is generated uniformly from $[1, \ldots, M]$.
>
> 3) Line 9: according to new line 8
>
> Let us consider how the proof changes in this case:
>
> 1) Lines 624-625:
>
> $$\mathbb{E_j} \left[\mathbb{E_Q} \left[\frac{1}{n} \sum \limits_{i=1}^n Q_i(F_{i, j_k^i}(m^k) - F_{1, j_k^1}(m^k) - F_{i, j_k^i}(u^k_H) + F_{1, j_k^1}(u^k_H))\right]\right] = F_{i}(m^k) - F_{1}(m^k) - F_{i}(u^k_H) + F_{1}(u^k_H))$$
>
> Here we use the unbiasedness of permutation compressors and of choices $j$ and their independence from previous points and iterations. As a result, the final result (up line 625) remains unchanged.
>
> 2) Lines 634-635:
> Here as a final result we get
> $$\frac{1}{n} \sum \limits_{i=1}^n || F_i (m^k) - F_1 (m^k) - F_i (u^k_H) + F_{1}(u^k_H) ||^2 \to \frac{1}{n} \sum \limits_{i=1}^n || F_{i, j_k^i}(m^k) - F_{1, j_k^1}(m^k) - F_{i, j_k^i}(u^k_H) + F_{1, j_k^1}(u^k_H)||^2$$
> We directly use Assumption 5.9 (new, defined here) and get (8) with $\delta \to \tilde \delta$.
>
> This means that in Lemma C.1 we have only $\delta \to \tilde \delta$.
>
> Lines 3-6 of Algorithm 1 also need to be adapted. They used deterministic ExtraGradient and now need to be modified for the stochastic case. Algorithm 1 from [1] is suitable for this. In the proof then we need to consider lines 647-651. In particular, the convergence of the deterministic ExtraGradient is used there now. It should be changed to the results of Theorem 4.9 and Corollary 4.10 of [1]. In particular, we have the following changes:
> $$
> \exp\left( - \frac{H}{4 \left(\gamma L + 1 \right)} \right) \to \exp\left( - \frac{H }{4 \sqrt{M} \left(\gamma L + 1 \right)} \right),
> $$
> $$
> \gamma = \mathcal{O}\left(\frac{H}{L}\right)  \to \gamma = \mathcal{O}\left(\frac{H}{\sqrt{M} L}\right)
> $$
>
> As a result of all changes, in the analogy of Corollary 5.3, we obtain the following estimate:
> $$
> \mathcal{O} (  [n + \frac{\tilde \delta \sqrt{n}}{\mu} + \frac{L \sqrt{M}}{\mu(H-1)}]\log \frac{1}{\varepsilon})
> $$
> With $H = 1 + \frac{L \sqrt{M}}{\tilde \delta \mu}$, we get
> $$
> \mathcal{O} (  [n + \frac{\tilde \delta \sqrt{n}}{\mu}]\log \frac{1}{\varepsilon})
> $$
> It turns out that the difference with the deterministic result is in $\delta$ (now we have $\tilde \delta$, which is larger than the original $\delta$) and in the number of local iterations of $H$.
>
> In a similar way, we can consider the case where we have stochasticity not of finite sum, but the following: we have access only to $\nabla_x f_i(x,y,\xi)$ and $\nabla_y f_i(x,y,\xi)$ and assume that $\mathbb{E}[\nabla_x f_i(x,y,\xi)] = \nabla_x f_i(x,y)$ and $\mathbb{E}[|| \nabla_x f_i(x,y,\xi) - \nabla_x f_i(x,y) ||^2] \leq \sigma^2$ (the same for $y$).

---

> > ### Comment · Reviewer_4qia · 2023-08-11
> > **Reviewer's response**
> >
> > Dear authors,
> >
> > thank you for this detailed rebuttal!
> >
> > All of my concern were addressed accordingly. I will maintain my original score.
> >
> > Best regards,

---

> > > ### Author Response · Authors · 2023-08-12
> > >
> > > Glad to hear that, thank you so much! Thanks again for your time spent on the review!

---

### Official Review · Reviewer_kubg · 2023-07-06

**Soundness:** 3 good
**Presentation:** 3 good
**Contribution:** 3 good
**Rating:** 6
**Confidence:** 4

**Summary:**

This paper studies distributed variational inequalities. It combines three techniques – similarity, compression, and local updates – to reduce the total number of communication rounds. It establishes theoretical guarantees of communication complexity.

**Strengths:**

1 It combines three techniques – similarity, compression, and local updates to reduce the total number of communication rounds for distributed variational inequalities.

2 It establishes tight communication complexity.


**Weaknesses:**

The tight bound on communication complexity relies on the assumption of data similarity. However, this assumption is not always valid, such as in the case of highly heterogeneous data.

**Questions:**

1 Which step of Algorithm 1 is the initialized $z_i^0$ used?

2 In line 186, it mentions "using only local information about operator $F_i$, changes the local value of variable $z$." In Algorithm 1, it seems that only the server updates $z$, and the device updates occur only in line 8 and line 13. There is no local $z_i$?

3 Line 676, it should be `` Proof of Theorem 5.7’’.


**Limitations:**

Yes.

---

> ### Author Rebuttal · Authors · 2023-08-08
>
> We thank Reviewer __kubg__ for the work, time, and positive feedback! Below we will respond to the issues raised.
>
> But before the response, we note that __we added new results to the new version of the paper - see our general response!__ Reviewer __JXTW__ requested __lower bounds. We made them and showed that our result is optimal and unimprovable!__ Reviewers __D8HM__ and __4qia__ asked to extend the results of the paper to __the stochastic case. This we also added!__ (see the response to Reviewer __4qia__) We hope that these results will further enhance the paper's contribution!
>
> __W__(data similarity):
>
> We agree that the assumption on data similarity is a bit limiting, yet it is popular in the literature. In the following, we cite papers from top conferences and journals where similarity was investigated [R1-11].
>
> [R1] Ohad Shamir, Nati Srebro, and Tong Zhang. Communication-efficient distributed optimization using an approximate newton-type method - __ICML__
>
> [R2] Yossi Arjevani and Ohad Shamir. Communication complexity of distributed convex learning and optimization - __NeurIPS__
>
> [R3] Yuchen Zhang and Xiao Lin. Disco: Distributed optimization for self-concordant empirical loss - __ICML__
>
> [R4] Xiao-Tong Yuan and Ping Li. On convergence of distributed approximate newton methods: Globalization, sharper bounds and beyond - __JMLR__
>
> [R5] Hadrien Hendrikx, Lin Xiao, Sebastien Bubeck, Francis Bach, and Laurent Massoulie. Statistically preconditioned accelerated gradient method for distributed optimization - __ICML__
>
> [R6] Sai Praneeth Karimireddy, Satyen Kale, Mehryar Mohri, Sashank J. Reddi, Sebastian U. Stich, and Ananda Theertha Suresh. SCAFFOLD: Stochastic controlled averaging for federated learning - __ICML__
>
> [R7] Amir Daneshmand, Gesualdo Scutari, Pavel Dvurechensky, and Alexander Gasnikov. Newton method over networks is fast up to the statistical precision - __ICML__
>
> [R8] Ye Tian, Gesualdo Scutari, Tianyu Cao, and Alexander Gasnikov. Acceleration in distributed optimization under similarity - __AISTATS__
>
> [R9] Aleksandr Beznosikov, Gesualdo Scutari, Alexander Rogozin, and Alexander Gasnikov. Distributed saddle-point problems under data similarity - __NeurIPS__
>
> [R10] Dmitry Kovalev, Aleksandr Beznosikov, Ekaterina Borodich, Alexander Gasnikov, and Gesualdo Scutari. Optimal gradient sliding and its application to distributed optimization under similarity - __NeurIPS__
>
> [R11] Ahmed Khaled, Chi Jin. Faster federated optimization under second-order similarity - __ICLR__
>
> __Q1__:
>
> We don't need them. Thanks! We removed them.
>
> __Q2__:
>
> In Section 4 (including line 186) we give general definitions without reference to Algorithm 1. In Section 5, we introduce Algorithm 1 and explain that local updates occur only on 1 device (paragraph "Local problem"). $u^k_H$ that comes from the server (line 7 of Algorithm 1) can be considered as $z_i$, but they are updated via communications, not local updates.
>
> __Q3__:
>
> Thanks! Fixed!

---

> > ### Comment · Reviewer_kubg · 2023-08-13
> >
> > I have read the rebuttal and I would like to keep the evaluation unchanged.

---

> > > ### Author Response · Authors · 2023-08-14
> > >
> > > Thank you so much! We really appreciate your work on our paper!

---

### Official Review · Reviewer_D8HM · 2023-07-07

**Soundness:** 2 fair
**Presentation:** 2 fair
**Contribution:** 2 fair
**Rating:** 6
**Confidence:** 3

**Summary:**

The paper proposes two novel distributed algorithms, for solving variational inequalities and saddle point problems. These algorithms combine three techniques - local updates, compression, and similarity - to address the communication bottleneck in distributed optimization for these problems. Algorithm 1 uses Tseng's method as the outer iteration and Extra Gradient for the local updates. It incorporates a compression technique based on variance reduction and permutation compressors that take advantage of data similarity. Algorithm 2 is an extension that uses partial participation to reduce local complexity.

**Strengths:**

The paper provides a thorough convergence analysis of the proposed algorithms under standard assumptions. The analysis derives competitive convergence rates and communication complexities that improve upon state-of-the-art methods. The analysis is also extended to the non-monotone case, demonstrating advantages over competitors. The theoretical work is a significant contribution. Comprehensive experiments on synthetic and real datasets confirm the benefits of the proposed algorithms over other baselines. The experiments show efficacy, especially as the degree of similarity between local functions increases. The results demonstrate both theoretical and practical significance.

**Weaknesses:**

Some concerns arise for this paper:

- The main concern I have for this paper is that, at each communication round only one client is calculating the gradients and updating the model. Hence, what is the number of communication required for convergence compared to other baselines? This has not been studied in the empirical section either.

- Also, there are other baselines in saddlepoint problems in federated learning, which this paper is missing to compare with, such as [A,B] with communication efficiency approaches.

- The algorithms and analysis are for the deterministic case. While the deterministic case is standard in the literature, the stochastic case may be more representative of machine learning applications. Analysis of the stochastic case would strengthen the work.



[A]Mohri, Mehryar, Gary Sivek, and Ananda Theertha Suresh. "Agnostic federated learning." International Conference on Machine Learning. PMLR, 2019.

[B]Deng, Yuyang, Mohammad Mahdi Kamani, and Mehrdad Mahdavi. "Distributionally robust federated averaging." Advances in neural information processing systems 33 (2020): 15111-15122.

**Questions:**

See above for questions.

**Limitations:**

They have discussed some of the limitations of their work in the paper, but it is more useful if we can have a detail explanation of them at the end.

---

> ### Author Rebuttal · Authors · 2023-08-08
>
> We thank Reviewer __D8HM__ for the time and the work! Next, we will answer the issues raised.
>
> But before the response, we note that __we added new results to the new version of the paper__ - see our general response! We added __the extension of our algorithms to the stochastic case__ (as requested by Reviewer __D8HM__ in Weaknesses) - see the response to Reviewer __4qia__. Reviewer __JXTW__ requested __lower bounds__. We made them and showed that __our result is optimal and unimprovable!__ We hope that these results will further enhance the paper's contribution!
>
> __W1__(communications):
>
> Unfortunately we don't fully understand the reviewer's concern, but we'll try to answer. Please clarify the question if we misunderstood the issue.
> 1) Algorithm 1 operates as follows. First (lines 2-6 of Algorithm 1), the server makes $H$ local iterations using only local information, communication is not needed here. Then (lines 7-8 of Algorithm 1), communication round occurs: the server sends parcels to all devices, and the devices in parallel send replies to the server. Thereafter (lines 9-10 of Algorithm 1), again only the local updates on the server take place (without communications). In lines 11-15 another communication round can also occur, but it happens with a small probability $p$. All of this is detailed in Sections 5.1 and 5.2. Section 5.2 also provides an estimate for the number of communication rounds required to achieve a solution with accuracy $\varepsilon$  (lines 286-293). Briefly, the number of communication rounds is $\mathcal{O}\left(\frac{\sqrt{n} \delta}{\mu} \log \frac{1}{\varepsilon}\right)$. Note that this is not the final result, since we compress the information by $n$ times during forwarding, we can assume that the total communication time cost is $n$ times less than number of rounds and is equal to $\mathcal{O}\left(\frac{\delta}{\sqrt{n} \mu} \log \frac{1}{\varepsilon}\right)$ (see Table 1). And this result on communication costs is optimal (can't be better) (we proved lower bounds - see the general response).
>
> 2) For Algorithm 2, we address partial participation. In this setting, only one device send response to the server during a communication round (see lines 8 and 9 in Algorithm 2, Appendix A). In comparison to Algorithm 1 (where compression is used and each device sends only a small portion of full vectoк), in Algorithm 2, only one device sends the whole vector; hence, the total amount of transmitted information is also $n$ times smaller than the number of communications rounds.  For partial participation our result on communication costs is also optimal (we proved lower bounds - see the general response).
>
> 3) Note that the estimate on communication rounds (not on communication cost) $\mathcal{O}\left(\frac{\sqrt{n} \delta}{\mu} \log \frac{1}{\varepsilon}\right)$ is worse than $\mathcal{O}\left(\frac{\delta}{\mu} \log \frac{1}{\varepsilon}\right)$ for the deterministic method (without compression) [16]. At the same time, our  total communication cost $\mathcal{O}\left(\frac{\delta}{\sqrt{n} \mu} \log \frac{1}{\varepsilon}\right)$ is better than $\mathcal{O}\left(\frac{\delta}{\mu} \log \frac{1}{\varepsilon}\right)$ from [16]. This effect is not due to local steps, but to the use of compression, which introduces stochasticity. It is not the case that by sending significantly less we will not lose in the number of communication rounds, we will lose, but we will gain more in terms of communication cost. That is the goal - to win significantly more than we lose! This is effect is predictable and standard in the literature around compressions - see [R1] (and Table 1 inside) or [R2] (and Table 1 inside).
>
> [R1] Gorbunov, E., Burlachenko, K. P., Li, Z., & Richtárik, P.. MARINA: Faster non-convex distributed learning with compression. In _International Conference on Machine Learning_
>
> [R2] Richtárik, P., Sokolov, I., & Fatkhullin, I.. EF21: A new, simpler, theoretically better, and practically faster error feedback. _Advances in Neural Information Processing Systems_
>
> __W2__(papers):
>
> We mentioned these papers in the new version of our paper, but they don't seem completely relevant to us. Let us explain with the example of the paper [B]. In this paper, the authors consider a general distributed/federated _minimization_ problem: $\min_x \frac{1}{n} \sum_{i=1}^n f_i(x)$. Then, motivated by robustness issues, the authors rewrite this problem as a special saddle point problem: $\min_x \max_{\lambda} \frac{1}{n} \sum_{i=1}^n \lambda_i f_i(x)$. This is not a general saddle problem, but a narrow formulation. But we consider general distributed saddle: $\min_x \max_y \frac{1}{n} \sum_{i=1}^n f_i(x, y)$. The same situation with [A].
>
> To give a clear example of our explanation before let us consider one probably the most famous modern and popular saddle point problem - GAN [R1]. This problem can be formalized as $\min_x \max_y f (x, y)$, where $x$ are weights of discreminator, $y$ are weights of generator. If we divide the training data among the devices, the problem becomes distributed: $\min_x \max_y \frac{1}{n} \sum_{i=1}^n f_i(x, y)$. Can this problem be rewritten as $\min_x \max_{\lambda} \frac{1}{n} \sum_{i=1}^n \lambda_i f_i(x)$. No
>
> [R1] Goodfellow et al. Generative adversarial nets
>
> __W3__(stochastic case):
>
> We added this (see our general response), hopefully this is what the reviewer wanted.

---

> > ### Comment · Reviewer_D8HM · 2023-08-19
> >
> > Thanks to the authors for a detailed response. They addressed most of my concerns. Hence, I will raise my initial score.

---

> > > ### Author Response · Authors · 2023-08-21
> > >
> > > Thank you so much for the reply and for the review!

---

### Author Rebuttal · Authors · 2023-08-08

Reviewer __JXTW__ requested lower bounds. We made them and showed that our result is optimal and unimprovable. Reviewers __D8HM__ and __4qia__ asked to extend the results of the paper to the stochastic case. This we also added. Because of the character limit, we give sketches of results and proofs. We are ready to provide detailed reasoning on the required facts upon separate request.

__Lower bounds__

We will slightly modify Assumption 4.3. The new version follows from the old one in the original paper.

_Assumption 4.3(new)_
For all $i \in 1,\ldots n$ and $u, v \in Z$
$$
\frac{1}{n} \sum\limits_{j=1}^n || F_i(u) - F_j(u)- F_i(v) + F_j(v)||^2 \leq \delta^2 ||u - v||^2
$$

1) _Upper bounds._ Since we change the Assumption 4.3, we need to show that this does not violate the old proofs.

We use Ass. 4.3 in line 635 (here we exactly need Ass 4.3(new)). We also use in (9) (lines 635-636):
$$
|| F(u) - F_1(u) - F(v) + F_1(v) ||^2 \le \frac{1}{n} \sum_{j=1}^n || F_j(u) - F_1(u) - F_j(v) + F_1(v) ||^2 \le \delta^2 ||u - v||^2
$$
Finally, we use Ass 4.3 in lines 666. Here to use Ass 4.3(new) we need to use that $i_k$ is generated uniformly:
$$
\mathbb{E}[ || F_{i^k}(m^k) - F_1(m^k) - F_{i^k}(u^k) + F_1(u^k) ||^2 ] = \frac{1}{n} \sum_{j=1}^n || F_j(u) - F_1(u) - F_j(v) + F_1(v) ||^2 \le \delta^2 ||u - v||^2
$$
It gives that all current proofs remain correct.

2) _Proof of lower bounds for partial participation._

We will show lower estimates for partial participation (Sec. 5.3), we give idea (by analogy with partial part.) of lower bounds for compression at the end. In this setup, we have a random set of devices that can participate in each communication. The point of obtaining lower bounds is to give a "bad" problem such that any algorithm from some class will perform no better than the lower bound.

First, we describe the class of algorithms for which we obtain lower bounds. The general idea of definitions can be found in [4, 16], the essence is that each device has its own output, due to local computations this output can be changed, the outputs can also be changed by communications with other devices. The starting output is 0. The final output is the server's output. The only thing that needs to be specified is the communication process (since we consider partial participation):

_At each communication round the server can sample uniformly and independently batch $S$ from [1,\ldots, n-1] of any size $b \in 1\ldots n-1$ and communicate with devices with numbers from $S$. Batch of the size $n-1$ is equal to the communication with all devices._

Next, we move to the "bad" problem. We consider saddle point problem (since it is special case of VIs)
$$
    f(x,y) = \frac{\delta}{4\sqrt{n}} x^T A y + \frac{\mu}{2}||x||^2 - \frac{\mu}{2}||y||^2 +  \frac{\delta^2}{2n\mu} e_1^T y,
$$
with
$$
A = \left(
\begin{array}{cccccccc}
1&-1 & & & & & &  \\\\
&1 &-1 & & & & &  \\\\
& & & &\ldots &\ldots & & \\\\
& & &   & & &1 &-1 \\\\
& & &  & & & &1 \\\\
\end{array}
\right)
$$
Construct functions $f_{i}$. Let define $a_q$ -$q$th row of the matrix $A$, $e_q$ - $q$th basis vector. We split function $f$ to sum $\frac{1}{n}\sum_{i=1}^n f_i$:
$$
    f_1 (x,y) = \frac{\mu}{2}||x||^2 - \frac{\mu}{2}||y||^2 + \frac{\delta^2}{8n\mu}e_1^T y
$$
for $i \in 2 \ldots n$:
$$
    f_i (x,y) = \sqrt{n-1} \cdot \frac{\delta}{4} x^T \left[ \sum\limits_{j \equiv (i-1) \text{mod} (n-1)} e_j a_j^T \right]y + \frac{\mu}{2}||x||^2 - \frac{\mu}{2}||y||^2 + \frac{\delta^2}{2n\mu}e_1^T y
$$

_Lemma 1._ $f_i$ satisfies Ass. 4.3(new) with $\delta$; $f_i, f$ satisfies Ass. 4.2 with $\mu$.

_Lemma 2._
After $N$ communications, in expectation only the first $\lfloor \frac{N}{n} \rfloor$ coordinates of the global output can be non-zero while  the rest of the $d-\lfloor \frac{N}{n} \rfloor$ coordinates are strictly equal to zero.

The essence of the lower bounds in this case is classical - how the final output close to the real solution is measured in the number of non-zero coordinates in this output. On non-zero coordinates we can (in the best case) get a number corresponding to the real solution, but on zero coordinates we cannot. The point is that we broke the problem into pieces $f_i$ and without communication we cannot increase the number of non-zero coordinates. Only one $f_i$ can move us to new non-zero coordinate, and this $f_i$ still needs to be sampled in the batches $S$.

Further getting lower bounds is pure algebra. The initial function $f$ coincides with the function for deterministic non-distributed lower bounds from [R1], the number of non-zero coordinates we have already found, so further we can simply use the results of algebraic transformations from Sec. 3.1 and 3.2 of  [R1] and get the lower bounds:

$$
\Omega\left( \sqrt{n} \frac{\delta}{\mu} \log \frac{1}{\varepsilon}\right).
$$

These bounds for the number of communication rounds (not for the cost - since cost $n$ times cheaper) - they are the same as Corollary 5.5 (if $H = \frac{L}{\delta\sqrt{n}} + 1$). It means that our Algorithm 2 is optimal.

3) _Proof of lower bounds for compression._

The idea here is exactly the same, but we need to restrict the class of compressors to random sparsification operators only (during communication round each device send random coordinates). In this case, all devices communicate, but each device samples the coordinate batch $S_i$ for sending. So we need to wait when the right device $f_i$ in its batch of coordinates $S_i$ find the right coordinate to increase the number of non-zero coordinates in the output.

[R1] Zhang et al. On lower iteration complexity bounds for the saddle point problems.

__Stochastic setup__

We proved the stochastic case of our algorithm in the response to Reviewer __4qia__.

---

### Decision · Program_Chairs · 2023-09-21

**Decision:**

Accept (poster)

**Comment:**

Among the reviewers it is a general consensus that this is a solid paper that should appear in the proceedings.